# On the ice-nucleating potential of warm hydrometeors in mixed-phase clouds

Michael Krayer[1], Agathe Chouippe[1,a], Markus Uhlmann[1], Jan Dušek[2], and Thomas Leisner[3]

[1]Institute for Hydromechanics, Karlsruhe Institute of Technology (KIT), Karlsruhe, Germany
[2]ICube, Fluid Mechanics Group, Université de Strasbourg, Strasbourg, France
[3]Institute of Meteorology and Climate Research, Atmospheric Aerosol Research Department, Karlsruhe Institute of Technology (KIT), Eggenstein-Leopoldshafen, Germany
[a]now at: ICube, Fluid Mechanics Group, Université de Strasbourg, Strasbourg, France

**Correspondence:** Michael Krayer (michael.krayer@kit.edu)

**Abstract.** The question whether or not the presence of warm hydrometeors in clouds may play a significant role in the nucleation of new ice particles has been debated for several decades. While the early works of Fukuta and Lee (1986) and Baker (1991) indicated that it might be irrelevant, the more recent study of Prabhakaran et al. (2020) suggested otherwise. In this work, we attempt to quantify the ice-nucleating potential using high-fidelity flow simulation techniques around a single hydrometeor and use favorable considerations to upscale the effects to a collective of ice particles in clouds. While we find that ice nucleation may be significantly enhanced in the vicinity of a warm hydrometeor and that the affected volume of air is much larger than previously estimated, it is unlikely that this effect alone causes the rapid enhancement of ice nucleation observed in some types of clouds, mainly due to the low initial volumetric ice concentration. Furthermore, it is demonstrated that the excess nucleation rate does not primarily depend on the rate at which cloud volume is sampled by the meteors' wakes, but is rather limited by the exposure time of ice nucleating particles to the wake, which is estimated to be of the order of few microseconds. It is suggested to further investigate this phenomenon by tracking the trajectories of ice nucleating particles in order to obtain a parametrization which can be implemented into existing cloud models to investigate second-order effects such as ice enhancement after the onset of glaciation.

## 1 Introduction

The formation of hydrometeors in clouds is of great importance for the prediction of weather and cloud electrification, as well as for the hydrological cycle, and thus, eventually for the evolution of climate. However, despite its relevance and great research effort over the past decades, many aspects remain poorly understood. One such puzzle is the discrepancy between the concentration of ice particles and that of available ice nuclei (IN) in airborne observations by several orders of magnitude (Pruppacher and Klett, 2010) which has been observed for various cloud types (Koenig, 1963; Auer et al., 1969; Hobbs, 1969; Hobbs and Rangno, 1985; Mossop, 1985; Hogan et al., 2002). This phenomenon has been termed *ice enhancement* and various mechanisms which amplify primary ice nucleation have been proposed to explain the observed surplus in ice particles (Field et al., 2017) .

The most promising class of enhancement mechanisms is the so-called *secondary ice production* (SIP), where new ice is formed from preexisting ice particles. Commonly accepted SIP include rime-splintering (Hallett and Mossop, 1974), fragmentation of ice (Vardiman, 1978; Takahashi et al., 1995; Bacon et al., 1998) and freezing drops (Hobbs and Alkezweeny, 1968). Most of these mechanisms have been implemented into cloud models with explicit ice microphysics (see Field et al. (2017) for a recent overview), which, however, are not capable of satisfactorily explaining the large amount of ice particles in observations. Further mechanisms have been proposed in the past whose relative importance is still to be evaluated.

While studying the effect of supersaturation on primary ice formation, Gagin (1972) proposed several mechanisms which locally produce high values of supersaturation, and thus, regions of significantly enhanced nucleation activity. He suggested that freezing hydrometeors, which attain higher temperatures than the surrounding air due to the release of latent heat, may cause transient supersaturation by simultaneous evaporation and heat transfer and linked it to the observation of satellite drops which have previously been observed experimentally during the freezing of supercooled droplets by Dye and Hobbs (1968). This hypothesis was corroborated by Nix and Fukuta (1974), who investigated numerically the transient freezing process of an isolated droplet.

Another phenomenon which leads to localized supersaturation around hydrometeors is the riming process of ice particles in mixed-phase clouds with high liquid-water content (Gagin, 1972). Here, sufficiently large ice particles collect supercooled droplets from their surroundings, which then accumulate on its surface and subsequently freeze, leading to similar non-equilibrium conditions as observed for the freezing drop. The supersaturation around a riming ice particle has been investigated numerically by Fukuta and Lee (1986) assuming steady conditions. Indeed, they found that the air in the vicinity of a warm ice particle can be highly supersaturated, while the magnitude and spatial extent strongly increase with increasing difference in surface to ambient temperature. However, even at low cloud temperatures, the supersaturated regions do not extend far away from the hydrometeor in their simulations, which led to the conclusion of Rangno and Hobbs (1991) that the overall ice enhancement is likely to be negligible.

While the previously mentioned studies focused on the quantification of the supersaturation field, Baker (1991) attempted to quantify the actual effect on ice enhancement around warm hydrometeors. He concluded that, although significant ice enhancement factors appear to be possible, the affected air mass seems to be too small to substantially contribute to the explanation of the discrepancy between ice crystal and ice nucleus concentrations.

Recently, the potential of hail and rain to nucleate droplets has gained new attention when Prabhakaran et al. (2017) observed condensation in the wake of cold droplets in a moist convection apparatus. In their experiment, a pressurized mixture of sulfur hexafluoride/helium was used and operating conditions were chosen such that slight supersaturations led to homogeneous condensation, which could indeed be observed in the wake of larger falling droplets. However, their results can only be qualitatively transferred to Earth's atmosphere mainly due to the difference in primary nucleation mechanism. Moreover, the mechanism to create supersaturation is different from the previously discussed one in the sense that the droplets are colder than the surrounding air.

To overcome these shortcomings, Prabhakaran et al. (2020) conducted a similar laboratory experiment using moist air which has been seeded with aerosol particles (AP) as an operating medium to investigate heterogeneous nucleation in the wake of

a warm falling droplet. Again, significant nucleation was observed in the wake of the falling hydrometeor. While the extent of wake-induced nucleation has not been quantified, the affected volume of air appears to be significantly larger than the predictions of Fukuta and Lee (1986) and Baker (1991) were suggesting. This discrepancy might be related to the strong assumptions on the flow made in these early works, which neglect important features of the wake of falling objects that are nowadays accessible to numerical simulations (Johnson and Patel, 1999; Bouchet et al., 2006; Zhou and Dušek, 2015). In particular, it has been shown that both unsteadiness and vortical structures of the flow strongly affect the temperature and vapor concentration in the vicinity and far downstream of a falling sphere (Bagchi et al., 2001; de Stadler et al., 2014), and thus strongly affect the distribution of supersaturation around hydrometeors (Chouippe et al., 2019).

In Chouippe et al. (2019), we presented a framework for high-fidelity numerical simulations of heat and mass transfer around a falling ice particle which is not in thermal equilibrium with its surroundings. Even though the focus was predominantly set towards several methodological questions, it was shown qualitatively that the local supersaturation differs strongly from the simpler considerations of Fukuta and Lee (1986).

The present work aims to revisit the details of the supersaturation field around an idealized falling hydrometeor and to quantify the volume of air which is affected by the presence of hydrometeors. Furthermore, it is yet to be evaluated how significant meteor-induced enhancement of ice nucleation is in clouds (Korolev et al., 2020), and therefore, we attempt to link our results to heterogeneous nucleation of aerosol particles.

## 2   Methodology

In order to assess the spatial structure of supersaturation around hydrometeors, knowledge on the flow around it is essential. This is not an easy task, since the features of the flow may strongly depend on parameters such as the size, shape and surface properties of the hydrometeor, which typically vary substantially, especially if ice particles are concerned.

In this work we utilize the numerical framework previously presented in Chouippe et al. (2019) and model the hydrometeor as a sphere with constant diameter $D$, constant surface temperature $T_p$ and a constant vapor pressure $e_{v,p}$ at its surface. The latter is determined by the assumption of local equilibrium at the surface, and thus, corresponds to the saturation vapor pressure $e_{sat,j}$ at $T_p$. Here, $j$ denotes the phase of water on the surface of the hydrometeor, where subscript "$i$" (ice) will be used to denote the solid phase and "$w$" (water) to denote the liquid phase.

The boundary conditions for the heat and mass transfer problem are motivated by the conditions expected for riming ice particles in the wet-growth regime (Ludlam limit). A surface temperature of $T_p = 0\,°\mathrm{C}$ is assumed, i.e. the surface of the ice sphere is warmer than the local cloud temperature, since the latter is generally below the freezing point in mixed-phase clouds. Please note that at this surface temperature the equilibrium vapor pressure with respect to both the liquid and solid phase coincide, and hence, no assumptions on the phase at the surface have to be made. Since the actual riming process is not modelled explicitly, but only enters through the boundary conditions of the heat and mass transfer problem, the results of this work are more general and can in principle be applied to any configuration which contains a mechanism leading to a warm

hydrometeor, such as a freezing droplet or rapid decreases in cloud temperature. Nonetheless, riming ice particles are likely the most relevant application, since the riming mechanism allows for a temperature gap which is sustained and not transient.

One of the key parameters for the investigation of wake-induced ice nucleation is the distribution of the saturation ratio

$$S_j(e,T) = e/e_{sat,j}(T), \tag{1}$$

where $e$ denotes the local vapor pressure. A parametrization of $e_{sat,j}$ with respect to temperature for both phase equilibria in the temperature range of interest is adopted from Murphy and Koop (2005). The spatial distribution of $S_j$ can then be reconstructed from the temperature field $T$ and the vapor concentration $n_v$, which are governed by the transport equations

$$\frac{\partial T}{\partial t} + \boldsymbol{u} \cdot \boldsymbol{\nabla} T = \mathcal{D}_T \boldsymbol{\nabla}^2 T, \tag{2}$$

$$\frac{\partial n_v}{\partial t} + \boldsymbol{u} \cdot \boldsymbol{\nabla} n_v = \mathcal{D}_{n_v} \boldsymbol{\nabla}^2 n_v, \tag{3}$$

where $\mathcal{D}_T$ and $\mathcal{D}_{n_v}$ denote the heat and vapor diffusivities and $\boldsymbol{u} = (u,v,w)^T$ is the velocity field of the surrounding air. The partial pressure of water is linked to these transported quantities by

$$e = n_v k_b T, \tag{4}$$

where $k_b$ is Boltzmann's constant. Even though the diffusivities of both vapor and heat are similar in magnitude, it was shown by Chouippe et al. (2019) that assuming equal diffusivities leads to an underestimation of the saturation ratio, and hence, both equations will be treated separately.

Under the boundary conditions described earlier, Eq. (2) leads to an outward heat flux, because the surface temperature of the hydrometeor is higher than that of its surrounding. Furthermore, at $T_p = 0\,°C$ the vapor concentration at the meteor's surface is generally higher than in the ambient, since the latter is assumed to be in equilibrium with the liquid phase at the corresponding cloud temperature $T_\infty < T_p$ due to the presence of droplets in mixed-phase clouds, i.e. $e_\infty = e_{sat,w}(T_\infty)$. This results in an outward vapor flux, and hence, the hydrometeor will also evaporate. It is therefore of primary importance to resolve the temporal and spatial variations of both temperature and concentration fields.

In Chouippe et al. (2019) we have shown that buoyant forces due to density variations within the fluid phase caused by variations in temperature and concentration are negligible at the parameter point of interest. The flow can then be approximated by the incompressible Navier-Stokes equations,

$$\boldsymbol{\nabla} \cdot \boldsymbol{u} = 0, \tag{5}$$

$$\frac{\partial \boldsymbol{u}}{\partial t} + (\boldsymbol{u} \cdot \boldsymbol{\nabla})\boldsymbol{u} = -\frac{1}{\rho_f}\boldsymbol{\nabla} p + \nu \boldsymbol{\nabla}^2 \boldsymbol{u}, \tag{6}$$

where $p$ denotes the hydrodynamic pressure, $\rho_f$ the density and $\nu$ the kinematic viscosity of the fluid. The flow is driven by the falling motion of the hydrometer, which is assumed to be moving with a constant velocity $\boldsymbol{v}_p$ equal to its terminal velocity $v_T$, i.e.

$$\boldsymbol{v}_p = -v_T \boldsymbol{e}_z, \tag{7}$$

where $e_z$ is the unit vector in the $z$-direction. The fluid ahead of the hydrometeor is assumed to be at rest. This setup is equivalent to a system with a fixed sphere positioned in an upcoming flow, which differs from that of a freely falling mobile sphere in the sense that fluctuations in $v_p$ are not permitted. The necessity to account for these variations has been discussed in Chouippe et al. (2019) and it was found that it has little influence in the present context due to the expected high value of the density ratio.

The Navier-Stokes and scalar transport equations are solved numerically in non-dimensional form. In particular lengths are scaled by the particle diameter $D$, velocity components by the terminal velocity $v_T$ and the scalar transport equations are formulated in terms of

$$\tilde{T} = (T - T_\infty)/(T_p - T_\infty), \tag{8}$$

$$\tilde{n}_v = (n_v - n_{v,\infty})/(n_{v,p} - n_{v,\infty}), \tag{9}$$

which has the advantage, that various dimensional boundary conditions, e.g. various values of $T_\infty$, can be studied in post-processing from a single simulation run. Under the simplifications stated, the non-dimensional problem can be parametrized by the Reynolds number, $Re = |v_p| D/\nu$, the Prandtl number, $Pr = \nu/\mathcal{D}_T$, and the Schmidt number, $Sc = \nu/\mathcal{D}_{n_v}$. While the Reynolds number is varied up to a value of $Re = 600$, the Prandtl and Schmidt numbers are set, respectively, to values of $Pr = 0.72$ and $Sc = 0.63$ in this study, which correspond to representative values expected for humid air in the temperature range of interest.

The numerical method employed is based on the method of Jenny and Dušek (2004), which has been used by e.g. Kotouč et al. (2008) and Chouippe et al. (2019) to simulate heat and mass transfer around spherical particles. It relies on a spectral/spectral-element discretiztion of the Navier-Stokes equations (Eq. 5,6) coupled to the transport of heat and mass (Eq. 2,3). The spherical particle is placed at the origin of a cylindrical domain $\Omega$. On the surface of the particle impermeability and no-slip boundary conditions are imposed, while the scalar fields are subject to the constant Dirichlet boundary conditions $\tilde{T} = \tilde{n}_v = 1$. At the upstream boundary of the simulation domain, a uniform velocity condition and $\tilde{T} = \tilde{n}_v = 0$ are enforced, while the downstream boundary is subject to zero-gradient boundary conditions for velocity as well as the scalars. The lateral boundaries are stress-free with zero-gradient boundary conditions for the scalars and zero pressure is imposed. The axial-radial plane is decomposed using the spectral element method of Patera (1984), while the homogeneous azimuthal direction is treated using Fourier decomposition. The numerical mesh used in this work is shown in fig. 1. For the temporal integration a time splitting method is used which consists of an explicit third order Adams-Bashforth discretization for the advective terms and a first order fully implicit discretization of the diffusion terms (Rønquist, 1988).

In the vertical direction the domain has a total length of $62D$, where the inflow length is $12D$ and the outflow length $50D$ measured from the center of the sphere. The diameter of the domain is $16D$. The computational domain has therefore been extended in the rear of the sphere compared to our former simulations presented in Chouippe et al. (2019). In total, 463 two-dimensional elements, each containing $[6 \times 6]$ collocation points, have been distributed over the domain. The azimuthal Fourier series is truncated at the 7th mode. This resolution is comparable to our previous work and has been shown to give good results for the momentum, as well has heat and mass transfer. All relevant scales of the flow and the scalar fields are resolved.

For more details on the numerical framework, the reader is referred to the previous work of Chouippe et al. (2019).

## 3 Results

### 3.1 Hydrometeor wake regimes in clouds

The dynamics of the flow around spherical objects are, under the assumptions stated in the previous section, fully parametrized by the Reynolds number, which depends on the diameter and indirectly, through modulation of the terminal velocity, on the density of the hydrometeor. Depending on the Reynolds number value, various flow states emerge in the wake (Johnson and Patel, 1999; Jenny et al., 2004; Kotouč et al., 2009). At low values the wake is axisymmetric and steady. When the first critical value of the Reynolds number, $Re_{c,1} \approx 212$, is exceeded, the wake becomes oblique with respect to the falling direction (*steady oblique regime)* and only planar symmetry may be observed (Ghidersa and Dušek, 2000). At $Re_{c,2} \approx 273$ a second instability of Hopf type occurs (Ghidersa and Dušek, 2000), which leads to unsteady periodic vortex shedding, while planar symmetry is still maintained (*oscillating oblique regime*). Eventually the vortex shedding becomes chaotic at $Re_{c,3} \approx 360$ (Ormières and Provansal, 1999) and all instantaneous symmetries are lost (*chaotic regime*). The listed regimes differ significantly in their ability to transfer heat and mass (Chouippe et al., 2019), and thus, different characteristics in producing local supersaturation are to be expected.

In order to estimate the distribution of ice particle wake regimes in clouds, we adopt the size distribution of Marshall and Palmer (1948),

$$\bar{N}_{met} = \bar{N}_{met,0} \exp(-\lambda D), \tag{10}$$

where $\bar{N}_{met}$ is the number concentration density of hydrometeors per unit volume of air and $\bar{N}_{met,0}, \lambda$ are model parameters. Equation (10) has originally been developed for raindrops, however, its validity for sufficiently large ice particles has been demonstrated by Passarelli (1978); Houze et al. (1979); Gordon and Marwitz (1984); Herzegh and Hobbs (1985); Patade et al. (2015) for various types of natural clouds. The values of the model parameters $\bar{N}_{met,0}, \lambda$ are usually obtained by airborne measurements and given as functions of the cloud temperature. One such parametrization is provided by Houze et al. (1979) for frontal clouds within a temperature range from $-42°C$ to $+6°C$, which will be used in the following for the size distribution of primary ice and is given by the constitutive equations

$$\bar{N}_{met,0}(T) = 5.5 \cdot 10^6 \exp(-0.09T)\,\mathrm{m}^{-4}, \tag{11}$$

$$\lambda(T) = 9.6 \cdot 10^{-1} \exp(-0.056T)\,\mathrm{mm}^{-1}, \tag{12}$$

where $T$ denotes the cloud temperature in °C. For this cloud type, ice enhancement has been previously reported in the presence of riming ice particles (Hogan et al., 2002). Even though it is known that the number density of ice particles of sub-millimeter size might deviate significantly from Eq. (10) in a sub-exponential or super-exponential manner (Passarelli, 1978), depending on cloud conditions, we assume that Eq. (10) holds nonetheless for all ice particle sizes for the sake of simplicity.

For reasons explained in section 3.3, our quantity of interest is not the number concentration, but rather the volume fraction $\phi$ of hydrometeors in clouds, which can be obtained by integrating the corresponding moment of the Marshall-Palmer distribution, i.e.

$$\phi = \int\limits_0^\infty \frac{\pi D^3}{6} \bar{N}_{met}(D)\,\mathrm{d}D. \tag{13}$$

In order to determine the wake regimes from the size distribution, the terminal velocity needs to be approximated as a function of the ice particle diameter, and hence, further assumptions on the density of the ice particles have to be made. For graupel particles, the density may take values which range between $0.05\,\mathrm{g\,cm^{-3}}$ and $0.89\,\mathrm{g\,cm^{-3}}$, depending on growth conditions and history (Pruppacher and Klett, 2010). For our estimation we choose a value of $0.6\,\mathrm{g\,cm^{-3}}$, as we are primarily concerned with ice particles in the wet-growth mode, which typically constitute the upper end of the density range. Using this value and the empirical drag law of Schiller and Naumann (1933), the critical diameters for regime transition are calculated from the critical Reynolds numbers assuming a fluid density of $1\,\mathrm{kg\,m^{-3}}$ and a kinematic viscosity of $1.68 \cdot 10^{-5}\,\mathrm{m^2\,s^{-1}}$, which corresponds approximately to the values expected at $750\mathrm{kPa}$ air pressure and $-10°\mathrm{C}$ ambient temperature (see table 1). The total solid volume fraction can then be subdivided into the contributions by ice particles with a certain wake regime, i.e.

$$\phi \equiv \sum_j \phi_j = \sum_j \int\limits_{D_{j,\mathrm{min}}}^{D_{j,\mathrm{max}}} \frac{\pi D^3}{6} \bar{N}_{met}(D)\,\mathrm{d}D, \tag{14}$$

where the index $j$ denotes one of the four wake regimes (steady axisymmetric; steady oblique; oscillating oblique; chaotic) and $D_{j,\mathrm{min}}, D_{j,\mathrm{max}}$ the corresponding lower/upper limit for the hydrometeor diameter to be in this regime.

| regime | axisymmetric | steady oblique | oscillating oblique | chaotic |
|---|---|---|---|---|
| $Re_c$ | 212 | 273 | 360 | – |
| $D_{j,\mathrm{min}}$ | 0 | $1.08\,\mathrm{mm}$ | $1.24\,\mathrm{mm}$ | $1.44\,\mathrm{mm}$ |
| $D_{j,\mathrm{max}}$ | $1.08\,\mathrm{mm}$ | $1.24\,\mathrm{mm}$ | $1.44\,\mathrm{mm}$ | $\infty$ |

**Table 1.** Summary of the critical Reynolds numbers for regime transition and the threshold diameters used to determine the partial volume fractions.

Figure 2 shows the estimated volume fraction of frozen hydrometeors in clouds as a function of cloud temperature. The values are typically smaller than $10^{-5}$ and decrease exponentially with decreasing temperature. When comparing the various regimes, it can be observed that the largest contributions generally stem from hydrometeors in the axisymmetric or chaotic regime. This becomes especially clear when looking at the relative contribution of $\phi_j$ to the total volume fraction $\phi$ as shown in fig. 3. As can be seen, more than $80\%$ of ice particles (by volume) exhibit one of these two wake regimes. Chaotic wakes are dominant at temperatures close to the freezing point, while axisymmetric wakes dominate when the temperature is very low. We will therefore focus solely on the axisymmetric and chaotic regimes in the following.

## 3.2 Supersaturation in the wake of hydrometeors

We now examine the saturation profiles in the wake of hydrometeors in the two regimes of interest. Figure 4 shows isocontours of supersaturation w.r.t. ice, defined as $s_i = S_i - 1$, for two different values of the Reynolds number, namely $Re = 75$, which lies within the axisymmetric regime, and $Re = 600$ in the chaotic wake regime. The ambient temperature was set to a value of $T_\infty = -15°C$, i.e. the temperature gap between the ambient air and the meteor surface is 15K, which corresponds to rather extreme cloud conditions with high liquid water contents (Greenan and List, 1995). Since the ambient fluid is already supersaturated w.r.t. ice, the threshold of the isocontours is given as an excess to the value in the ambient. At $Re = 75$, the flow is steady, and thus, so is the supersaturation field. In the chaotic regime, the flow is characterized by time-dependent vortex shedding from the ice particle's boundary layer, and therefore excess supersaturation appears intermittently. In both regimes, significant excess supersaturation w.r.t. the ambient can be observed far downstream and the volume of air which is affected by the wake is by far larger than the volume of the meteor. This can be seen more clearly when averaging the fields in the azimuthal direction, which is statistically homogeneous for both cases, as well as in time for the unsteady flow. The averaged excess supersaturation w.r.t. ice is shown in fig. 5 for both cases. The saturation profiles differ substantially from the ones obtained by Fukuta and Lee (1986) for similar boundary conditions and Reynolds number. This is presumably due to their strong simplification of potential flow, which is incapable of reproducing the boundary layer and the flow in the wake of the ice particle correctly. This indeed leads to strong modifications in the distribution of supersaturated regions. Regions of high vapor content and relatively low temperatures, i.e. supersaturated regions, which are created in the mixing layer close to the riming particle are transported further downstream by the detaching vortices in the chaotic regime. This phenomenon does not occur in the simulations of Fukuta and Lee (1986), who observed a complete decay of supersaturation to the inflow value within few meteor diameters downstream (approximately within two diameters at $-20°C$), while fig. 5 clearly indicates that regions with $s_i(\boldsymbol{x}, t) > s_{i,\infty}$ may be observed at downstream distances of the order of $50D$ at $-15°C$.

In order to compare the induced supersaturation for different values of $T_\infty$ and in between different regimes, we compute the time-averaged volume of air which is supersaturated above a given threshold,

$$V_s(s^*) \equiv \int_\Omega \langle H(s(\boldsymbol{x}, t) - s^*) \rangle_t \, d\boldsymbol{x}, \tag{15}$$

where $V_s$ is the superaturated volume, $s^*$ the corresponding threshold and $H$ the Heaviside step function. The integral is to be understood as a volume integral over the simulation domain $\Omega$ and $\langle . \rangle_t$ denotes the time-averaging operator. The supersaturated volume can be computed w.r.t. ice or the liquid phase which will be indicated in the following by a phase subscript for the threshold variable.

Figure 6 shows the normalized volume of air superaturated w.r.t. ice (a) and liquid (b) as a function of the thresholds $s_i^*$ and $s_w^*$, respectively. It can be seen that in the axisymmetric regime, the supersaturated volume is generally larger than in the chaotic regime. This is caused by the enhanced mixing properties of the vortical wake structures in the latter case, which lead to a faster decay of the scalar field. For both regimes, the range of observed values of supersaturation is similar. In fact, differences in range may only occur due to the distinct diffusivities of the temperature and water vapor fields, which indeed

lead to subsaturations in the chaotic regime in some parts of the flow. While the range can be estimated reasonably well using simple mixing arguments (Chouippe et al., 2019), we observe that the highest values of supersaturation only occur in a very confined portion of the wake. However, if the temperature difference between the ambient and the ice particle is sufficiently high, a strong excess in supersaturation compared to the ambient value may occur for both phases in a volume comparable to the size of the meteor, if no mechanism for depletion is considered. The volume of air which is affected by the wake-induced supersaturation is of the order of $10^3$ particle volumes for all temperature gaps of interest.

### 3.3 Ice enhancement due to meteor wakes

In the previous section it was shown that considerable supersaturations may be observed in the wake of hydrometeors and that the volume of air which is affected by it is far larger than what has been previously estimated in the literature. In order to analyze our spatially resolved data in terms of ice nucleation, a constitutive relation is required which links supersaturation to the concentration of activated ice nuclei (IN). A possible parametrization obtained from continuous-flow diffusion chamber (CFDC) measurements for natural aerosols is given by Meyers et al. (1992) and reads

$$N_{IN} = 0.528 \exp\left(12.96 s_i\right) \mathrm{m}^{-3}. \tag{16}$$

Here, $N_{IN}$ denotes the number concentration of IN activated by the deposition and condensation-freezing mechanisms. Eq. (16) is reported to be strictly valid in the parameter range

$$-20°\mathrm{C} < T < -7°\mathrm{C}, \quad 0.02 < s_i < 0.25, \quad -0.05 < s_w < 0.045, \tag{17}$$

which corresponds to the range of the underlying CFDC data, albeit it has also been applied outside of this range (Meyers et al., 1992). The temperatures of interest in the current work fall well into the range of validity. As the distribution of $s_i$ in the wake (fig. 6a) shows, $s_i$ does only slightly exceed $25\%$ at the lowest ambient temperature considered. However, only a small portion of the wake exhibits such high values, and hence, imprecisions are likely to be insignificant for integral quantities. The volumetric distribution of $s_w$ (fig. 6b) indicates that supersaturation w.r.t. liquid exceeds the CFDC data range in significant portions of the wake when $T_\infty \lesssim -10°\mathrm{C}$. Therefore, the constitutive relation may underestimate the contribution of the condensation-freezing mode, as this mode exhibits increased activity under these conditions, as has been demonstrated by Schaller and Fukuta (1979) for various substances. Having this in mind, we are not aware of any parametrization of condensation-freezing nucleation for natural aerosols which can be directly applied under these conditions.

One way to quantify the enhancement of nucleation activity is the ice enhancement factor, i.e. the ratio between the observed number concentration of ice nuclei and that expected for a reference state. This concept has been used before by Baker (1991) to study meteor-induced ice enhancement. For our problem, the local ice enhancement factor can be expressed as

$$f_i(\boldsymbol{x}) = \frac{\langle N_{IN}(\boldsymbol{x},t) \rangle_t}{N_{IN,\infty}}, \tag{18}$$

where $N_{IN}(\boldsymbol{x},t)$ is calculated from the instantaneous local supersaturation w.r.t. ice using Eq. (16) and $N_{IN,\infty}$ is the IN concentration evaluated for ambient conditions. Thus, the ice enhancement factor gives a quantitative measure on how many additional IN are activated in the wake of the hydrometeor compared to the ambient.

Figure 7 visualizes the local ice enhancement factor in the wake of a hydrometeor at $T_\infty = -15°C$. At this rather favorable parameter point, we observe that ice nucleation is considerably increased (up to a factor of 6) in the near vicinity of the hydrometeor and significant values may still be observed in the far wake. It is therefore conceivable that a considerable amount of new ice may be formed in the wake, which is consistent with recent experimental results of Prabhakaran et al. (2020) who observed nucleation of water droplets and ice particles in the wake of a hot drop under cold conditions.

Using Eq. (16), the supersaturated volume can be expressed as a function of the ice enhancement factor in order to gain insight into the volumetric distribution of $f_i$. Figure 8 shows the distributions for various temperatures in both wake regimes. The ice enhancement is generally higher in the axisymmetric regime than in the chaotic regime. Please note, that for a more precise analysis of this behavior, the actual flow-driven distribution of AP in the wake should be considered. Depending on the flow dynamics, ice nucleating particles may accumulate in the wake or be expelled from it (Homann and Bec, 2015), which might have a non-negligible impact on ice enhancement. However, the consideration of aerosol dynamics is outside the scope of the present work.

The analysis of the local ice enhancement factor leads to the conclusion that the wake of a hydrometeor may act as a site of increased nucleation activity, and that the volume in which this increased activity occurs is much larger than the volume of the hydrometeor itself. Having identified wake-induced supersaturation as a possible ice multiplication mechanism in the near surrounding of a hydrometeor, its significance in the larger context of clouds is yet to be investigated. In order to assess this, we suggest to compute a coarse-grained ice enhancement factor, which takes into account the spatial distribution of $f_i$ obtained by the numerical simulations, as well as the concentration of primary ice particles. Therefore, we define a control volume $\mathcal{V}$ within a cloud which is sufficiently large such that the hydrometeor size distribution follows Eq. (10), but sufficiently small such that the saturation field is uniform (with a value of $s_{i,\infty}$) if no hydrometeor is present. The volume-averaged ice enhancement factor within this control volume is given by

$$\langle f_i \rangle_\mathcal{V} = \frac{\int_\mathcal{V} (\langle N_{IN}(\boldsymbol{x},t)\rangle_t - N_{IN,\infty})\, \mathrm{d}\boldsymbol{x}}{\int_\mathcal{V} N_{IN,\infty}\, \mathrm{d}\boldsymbol{x}} + 1, \tag{19}$$

where the integral has been decomposed into an excess contribution, which evaluates to zero outside of the wake, and a base contribution, which is equal to unity. At sufficiently small volume fractions, the wakes of individual hydrometeors can be assumed to not interact, and hence, the integral in the enumerator of the first term in Eq. (19) can be expressed as a sum of contributions from independent hydrometeors, i.e.

$$\int_\mathcal{V} (\langle N_{IN}(\boldsymbol{x},t)\rangle_t - N_{IN,\infty})\, \mathrm{d}\boldsymbol{x} = \sum_{(k)} \int_{\Omega^{(k)}} (\langle N_{IN}(\boldsymbol{x},t)\rangle_t - N_{IN,\infty})\, \mathrm{d}\boldsymbol{x}, \tag{20}$$

where $\Omega^{(k)} \subseteq \mathcal{V}$ indicates that the field $N_{IN}(\boldsymbol{x},t)$ is to be taken from a simulation with an appropriate value of the Reynolds number for meteor $(k)$. Assuming a continuous particle size distribution according to Eq. (10), the volume-averaged ice enhancement within $\mathcal{V}$ can be expressed as

$$\langle f_i \rangle_\mathcal{V} = 1 + \frac{1}{N_{IN,\infty}} \int_0^\infty \bar{N}_{met}(D) \int_{\Omega(D)} (\langle N_{IN}(\boldsymbol{x},t)\rangle_t - N_{IN,\infty})\, \mathrm{d}\boldsymbol{x}\, \mathrm{d}D, \tag{21}$$

where $\Omega(D)$ denotes the simulation domain for a meteor with diameter $D$. An evaluation of this expression requires knowledge on how the supersaturation field evolves with the Reynolds number. However, our results for the two regimes investigated indicate that the values of the non-dimensional integral

$$\tilde{I}_{\Omega(D)} \equiv \frac{6}{\pi} \int\limits_{\Omega(D)} (\langle N_{IN}(\tilde{\boldsymbol{x}},t)\rangle_t - N_{IN,\infty})\,\mathrm{d}\tilde{\boldsymbol{x}} \tag{22}$$

is of the same order of magnitude for a wide range of Reynolds number, as they only differ approximately by a factor of 3 for the two Reynolds numbers investigated. For reasons of simplicity, we therefore assume that $\tilde{I}_{\Omega(D)}$ can be approximated for a given regime by the value determined for a single Reynolds number within that regime, $\tilde{I}_{\Omega_j}$, which allows us to rewrite Eq. (21) in the simplified form

$$\langle f_i \rangle_{\mathcal{V}} = 1 + \frac{1}{N_{IN,\infty}} \sum_j \int\limits_{D_{j,\min}}^{D_{j,\max}} \frac{\pi D^3}{6} \bar{N}_{met}(D)\tilde{I}_{\Omega_j}\,\mathrm{d}D. \tag{23}$$

Since $\tilde{I}_{\Omega_j}$ is now a constant for each regime, this can be reexpressed using Eq. (14) to yield

$$\langle f_i \rangle_{\mathcal{V}} = 1 + \sum_j \phi_j \tilde{I}_{\Omega_j}/N_{IN,\infty}, \tag{24}$$

which demonstrates the importance of the ice volume fraction on global ice enhancement. As has already been discussed and shown in fig. 3, the axisymmetric and chaotic regimes contribute the most to the volume fraction. We therefore disregard the contribution of the two other regimes and adjust the threshold of regime transition accordingly to a value of $1.26\,\mathrm{mm}$, which allows us to evaluate Eq. (24) from our flow simulation data.

Both $\phi_j$ and $\tilde{I}_{\Omega_j}$ are functions of ambient temperature, and thus the global ice enhancement factor originating from meteor wakes can be expressed as a function of cloud temperature. While the former decreases exponentially with decreasing temperature, the latter exhibits a strong increase leading to a counteracting effect. Furthermore, a large number of ice particles pertain to the axisymmetric regime for low temperatures, which is slightly more favorable in terms of ice enhancement.

Please be aware that, even though our aim is to quantify the wake-induced ice formation, the volume fraction of ice will be regarded as constant. The reason for this is that the approach used in this study does not allow us to derive the time scales of growth for newly created ice, and thus, the time-dependent coupling with the size distribution is inaccessible. Therefore, the following considerations merely apply to the initial state of a possible rapid glaciation process.

Figure 9 shows the global ice enhancement factor as a function of ambient temperature. It can be seen that ice nucleation in a given cloud volume is only marginally increased due to the presence of meteor wakes. At the highest temperature gap considered, the induced supersaturation activates roughly $0.08\%$ more ice than the ambient, and hence, we conclude that under typical cloud conditions hydrometeor wakes do not considerably affect ice nucleation on a cloud scale, at least when the volume fraction of ice in the cloud is as low as predicted by the parametrization of Houze et al. (1979).

Since the volume fraction is the limiting factor for ice enhancement, we revisited our assumption on the size distribution of ice particles in order to assess the sensitivity to the parametrization. Using the parametrization of Heymsfield et al. (2002) for

deep subtropical and tropical clouds, the ice volume fraction is of the same order of magnitude as for the presented results, such that the conclusions which have been drawn for frontal clouds also persist for this type of cloud.

### 3.4 Time scales of nucleation and history effects

The ice enhancement analysis presented in the previous subsection did not explicitly address two factors which may be of importance when investigating IN activation, namely the time of exposure required to actually activate an AP and history effects, i.e. the circumstance that IN which have been activated in the wake may stay activated once they are not exposed to the wake anymore. The latter leads to an accumulative effect, which suggests that it is more reasonable to consider the

340 volume swept by the meteors rather then looking at the volume within a cloud which is instantaneously exposed to high supersaturations. Indeed, Prabhakaran et al. (2020) used this argument to show that essentially all interstitial AP within a cloud volume of interest are exposed to high supersaturations within a few minutes. Following their arguments, we estimated the time required to expose a significant portion of the cloud to the wakes using

$$\tau_{sweep} = \left( \int\limits_0^\infty \bar{N}_{met}(D) \dot{V}_{sweep}(D) \, \mathrm{d}D \right)^{-1}, \tag{25}$$

where $\dot{V}_{sweep} = \epsilon v_p D^2 \pi / 4$ is the volumetric flow rate of air which is swept by a hydrometeor with diameter $D$ and velocity $v_p$ and $\epsilon$ being an unknown factor assumed to be of the order of unity. Using the size distribution of Houze et al. (1979) and the terminal velocity for smooth spheres, we obtain $\tau_{sweep} \approx 110$s at $-15°$C, which fits the estimation of Prabhakaran et al. (2020) very well. The use of an empirical law for the terminal velocity of frozen hydrometeors of natural shape leads to larger values of $\tau_{sweep}$, which are, however, found to be of similar order of magnitude.

In the following we apply the swept-volume argument to ice multiplication by quantifying the excess ice nucleation rate induced by the meteor wakes, henceforth denoted as $j_{met}$. Under the assumption that nucleation occurs sufficiently fast to achieve the IN concentrations predicted by Eq. (16), the nucleation rate can be estimated from the number of IN activated in the wake and the time it takes to replenish the volume of fluid affected by high supersaturations. The former is obtained directly from our simulation data by computing the volume integral $\int_{\Omega(D)} (\langle N_{IN}(\boldsymbol{x},t) \rangle_t - N_{IN,\infty}) \, \mathrm{d}\boldsymbol{x}$ while the latter is difficult to

define objectively. We propose to estimate the time scale of wake renewal by

$$\tau_{expo} \equiv V_{aff} / \dot{V}_{sweep} \tag{26}$$

where $V_{aff} = \gamma D^3 \pi / 6$ is the volume affected by the wake of a hydrometeor of diameter $D$, which should be proportional to the volume of the hydrometeor. The prefactor $\gamma$ is currently unknown, but might be related to the concept of supersaturated volume. The time scale $\tau_{expo}$ can be interpreted as the characteristic time a fluid volume is exposed to high supersaturations, and hence the subscript. From the definitions of $V_{aff}$ and $\dot{V}_{sweep}$ it follows that $\tau_{expo} \propto D / v_p$ with the constant of proportionality being referred to as $C_{expo}$ hereafter. This new constant contains both unknown coefficients $\epsilon$ and $\gamma$ and might be understood as the non-dimensional streamwise length of the wake. Again, this length is difficult to define rigorously due to the asymptotic decay of supersaturation, but judging from fig. 7, it is likely of the order of 10. The swept-volume limited nucleation rate for

an ensemble of meteors is then given by

$$j_{met}^{expo} = \int\limits_0^\infty \bar{N}_{met}(D) \frac{1}{\tau_{expo}(D)} \int\limits_{\Omega(D)} (\langle N_{IN}(\boldsymbol{x},t) \rangle_t - N_{IN,\infty}) \, \mathrm{d}\boldsymbol{x} \, \mathrm{d}D \tag{27}$$

under the assumptions that IN activate sufficiently fast during the exposure, a sufficient number of activatable AP is present and that those are homogeneously distributed within the wake. The resulting nucleation rate is displayed in fig. 10 for $C_{expo} = 10$ (solid blue line) and the range $1 < C_{expo} < 100$ in order to pay regard to the uncertainties associated with this tunable parameter (shaded blue area). Under the assumptions just stated, meteor-wakes indeed appear to be capable of activating a large number of IN in a short amount of time, as has been concluded by Prabhakaran et al. (2020). In fact, at $-10°\mathrm{C}$ it would only take around 40s for the number concentration of wake-activated IN to match the concentration of primary meteors.

This result strongly contradicts our conclusion that wake-induced supersaturation is of little significance in clouds based on the ice enhancement factor. As will be demonstrated in the following, the resolution to this apparent contradiction requires an examination of the time scales of nucleation and exposure. For $C_{expo} = 10$, the exposure times are of the order of 5ms for all diameters of interest, which is substantially shorter than the time scales usually relevant for cloud modelling (of the order of minutes). As Eq. (16) has been developed for cloud modelling, the validity of the assumption that IN activation can be regarded as instantaneous at the time scales considered should be brought into question. Indeed, classical nucleation theory (Fletcher, 1958) suggests that nucleation is a time-dependent process until the activated fraction of AP approaches unity. From concentrations of IN obtained from continuous-flow diffusion chamber (CFDC) experiments, the nucleation rate may be estimated by taking into account the residence time in the apparatus, $\tau_{nucl}$ (Hoose and Möhler, 2012). Since Eq. (16) is based on CFDC data, we make the conjecture that the local nucleation rate may be approximated by the relationship

$$j_{IN}(\boldsymbol{x},t) \approx N_{IN}(\boldsymbol{x},t)/\tau_{nucl}. \tag{28}$$

In Hoose and Möhler (2012) residence times ranging from 1.6s to 120s are reported for various CFDC experiments. The primary data used to obtain Eq. (16) also suggests that the peak concentration $N_{IN}$ is achieved with residence times of approximately 10s (Al-Naimi and Saunders, 1985) and that shorter residence times lead to lower concentrations, which is in accordance to the arguments stated above. The crucial assumption of Eq. (27) that IN concentrations predicted by Eq. (16) can be achieved within the exposure time is disproved by acknowledging that $\tau_{nucl} \gg \tau_{expo}$. The large discrepancy in time scales suggests that the nucleation rate is not limited by the rate at which interstitial AP are entrained into the wake (the swept volume), but rather by the time scale of the nucleation process itself, and thus by the instantaneously exposed volume. A more appropriate estimate of the nucleation rate is then given by

$$j_{met}^{nucl} = \int\limits_0^\infty \bar{N}_{met}(D) \frac{1}{\tau_{nucl}} \int\limits_{\Omega(D)} (\langle N_{IN}(\boldsymbol{x},t) \rangle_t - N_{IN,\infty}) \, \mathrm{d}\boldsymbol{x} \, \mathrm{d}D \tag{29}$$

under the assumption that AP are entrained sufficiently fast into the wake, which seems reasonable given the arguments by Prabhakaran et al. (2020), and that they are distributed homogeneously within the wake. The new estimation of the nucleation

rate is shown in fig. 10 for $\tau_{nucl} = 10$s (solid red line) and the range $1$s $< \tau_{nucl} < 100$s (shaded red area). As can be seen, the rates are several orders of magnitude smaller than what has been previously estimated, and it would now take around 82h for the number concentration of wake-activated IN to match the concentration of primary meteors, which is in better agreement to the conclusion obtained with the help of the ice enhancement factor. In fact, it is straightforward to demonstrate that these two quantities are directly linked by the relationship

$$\langle f_i \rangle_{\mathcal{V}} = \frac{j_{met}^{nucl}}{N_{IN,\infty}/\tau_{nucl}} + 1. \tag{30}$$

Furthermore, it can be shown that as soon as a significant cloud volume is swept quickly at low volume fractions, the exposure time will automatically be short as $\tau_{expo} \propto \tau_{sweep}$ for a given meteor concentration, and thus, ice enhancement is again found to be limited by the low volumetric concentration of ice in clouds.

## 4  Conclusion

In this study we have performed numerical simulations of momentum, heat and mass transfer around a warm hydrometeor in order to assess the distribution of supersaturation in its wake, and moreover, the implications on ice nucleation enhancement. Our simulation method is based on a body-conforming spectral/spectral-element discretization and all relevant scales of the flow problem have been resolved. The hydrometeor is assumed to be of spherical shape and to possess a uniform surface temperature of $0\,^{\circ}$C, while the ambient temperature has been varied in a range between $-15\,^{\circ}$C and $0\,^{\circ}$C. The vapor concentration is kept at saturation value with respect to ice at the particle surface, while it is saturated with respect to water in the ambient (reflecting the presence of supercooled droplets). Two different values of the Reynolds number have been simulated in order to capture the characteristics of the most relevant wake regimes, namely $Re = 75$, where the wake is steady and axisymmetric, and $Re = 600$, where the wake is chaotic.

We found that significant values of supersaturation can be attained in the wake of warm hydrometeors, which persist long enough to be observed at several tens of particle diameters downstream of the meteor for sufficiently high differences in temperature. The supersaturated volume of air exceeds the estimations by Fukuta and Lee (1986) by far, which is attributed to the more accurate representation of the flow in the current study. This is an important observation since one of the key arguments for disregarding wake-induced ice nucleation is the proclaimed small zone of influence (Baker, 1991). It should be borne in mind that the ambient air was considered to be quiescent in this study. Under atmospheric conditions, turbulent fluctuations with scales comparable in size with the hydrometeors are expected to be present, which will presumably lead to a faster decay of the vapor concentration and temperature in the wake (Bagchi and Kottam, 2008). It could be worthwhile to investigate the influence of turbulence on the supersaturated volume in the future, as well as the importance of fluctuations in the ambient saturation ratios in comparison to wake-induced fluctuations.

Using a constitutive relation provided by Meyers et al. (1992) for the ice nuclei concentration, we estimated that heterogeneous nucleation may be locally enhanced by a factor of up to 6 if the temperature gap between the ambient and the meteor's surface is sufficiently large. This enhancement increases strongly with decreasing cloud temperature and so does the affected

volume of air, which is typically of the order of several hundred particle volumes. It is therefore conceivable that ice multiplication can be triggered by the meteor's wakes, which is in agreement with the recent experimental observations by Prabhakaran et al. (2020). However, as arguments on upscaling have shown, this local effect alone presumably has little significance in clouds due to the low volumetric concentration of primary ice. This conclusion appears contradictory to the arguments given

by Prabhakaran et al. (2020), who assessed that hydrometeors are capable of sampling a large cloud volume with their wakes within a short amount of time. While this is indeed the case, we have shown that the time during which ice nucleating particles are exposed to the wake is generally too short to activate all of them, even at high values of supersaturation. The nucleation rate is therefore not limited by the rate at which interstitial AP are encountered, and therefore, the instantaneously exposed volume is argued to be more relevant, supporting the results obtained with the help of the ice enhancement factor.

In order for wake-induced supersaturation to be a relevant SIP, the nucleation rate needs to be considerably higher than what has been estimated in this work. While it might be possible that the condensation-freezing mode has been underestimated in this work, it seems unlikely that this underestimation is significant enough to substantially affect the conclusion. As the relevance of this SIP is mainly determined by the volumetric concentration of ice particles, high contents of ice are necessary for it to be active. Furthermore, in order to achieve an adequate temperature difference between the ambient and the ice, high liquid water

contents are also required. Hence, if this SIP occurs in natural clouds, convective clouds are the most likely candidates.

Nonetheless, it is conceivable that the present mechanism in conjunction with one (or several) secondary ice processes is of greater relevance to the problem of ice formation. After a rapid glaciation process has been triggered, wake-induced nucleation might become significant, as the ice concentration then increases to considerable values. Futhermore, the question of whether or not wake-induced nucleation alone might trigger such a process has not been fully resolved yet, since the feedback on the

445 size distribution of ice particles has been disregarded in this study. In order to assess this dynamical behavior, further time-resolved information on the activation of ice nuclei in the wake appears to be necessary. By tracking the trajectories of AP, the estimation of the nucleation rate may be further improved and a resulting parametrization may be added to existing cloud models with explicit microphysics in order to assess the relative importance of wake-induced nucleation in comparison to other secondary ice processes.

*Data availability.* The datasets are available upon request to the corresponding author.

*Author contributions.* TL and MU conceptualized the idea. JD provided the simulation code and mesh. MK conducted the simulations. MK, AC and MU post-processed the data and TL assisted in the interpretation of the results. MK prepared and revised the manuscript and AC contributed to it. MU, AC, JD and TL proofread the manuscript.

*Competing interests.* The authors declare that they have no conflict of interest.

*Acknowledgements.* The authors thank Alexei Kiselev for helpful discussions. The constructive feedback by Alexei Korolev and an anonymous reviewer is greatly acknowledged as their suggestions helped to improve and clarify this manuscript. The simulations were partially performed at the Steinbuch Centre for Computing in Karlsruhe and the computer resources, technical expertise and assistance provided by this center are thankfully acknowledged. This article has been funded through the Open Access Publishing Fund of Karlsruhe Institute of Technology.

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

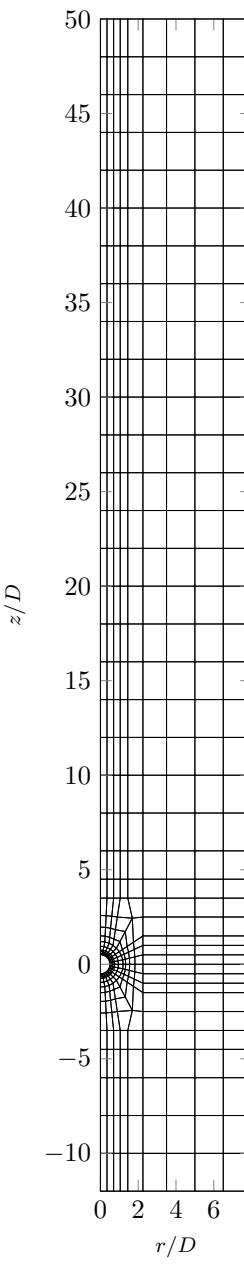

**Figure 1.** Spectral-element mesh in the axial-radial plane, where the lines depict the boundaries of the 463 elements. Each element contains $[6 \times 6]$ collocation points. The three-dimensionality is introduced by Fourier expansion in the azimuthal direction, which is truncated at the 7th Fourier mode, resulting in a cylindrical domain. A uniform velocity profile with constant temperature and vapor content is imposed at the upstream boundary, while the downstream boundary is subject to zero-gradient BCs for both velocity and scalars. The lateral boundaries are stress-free at zero pressure with zero-gradient BCs for the scalars. The surface of the spherical particle, whose center is located at the origin of the coordinate system, is impermeable and a no-slip BC for velocity is imposed as well as constant Dirichlet BCs for the scalar fields.

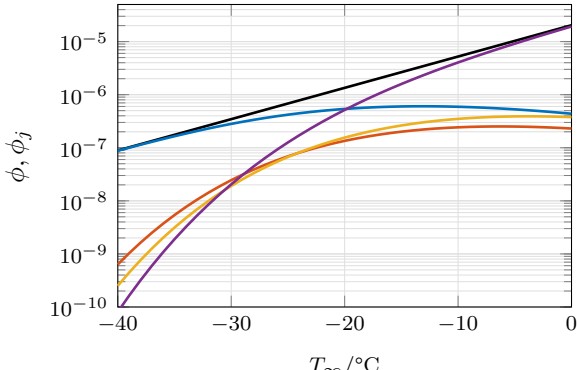

**Figure 2.** Volume fraction of ice particles in clouds as a function of ambient temperature. The total volume fraction $\phi$ is displayed by a solid black line (———), while the contribution by the regimes is given by the colored lines. Linestyles: axisymmetric regime (———), steady oblique regime (———), oscillating oblique regime (———), chaotic regime (———).

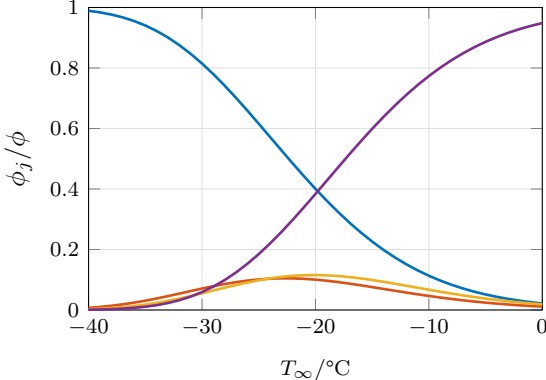

**Figure 3.** Relative contribution to the total volume fraction by hydrometeors in a certain regime. Linestyles: axisymmetric regime (———), steady oblique regime (———), oscillating oblique regime (———), chaotic regime (———).

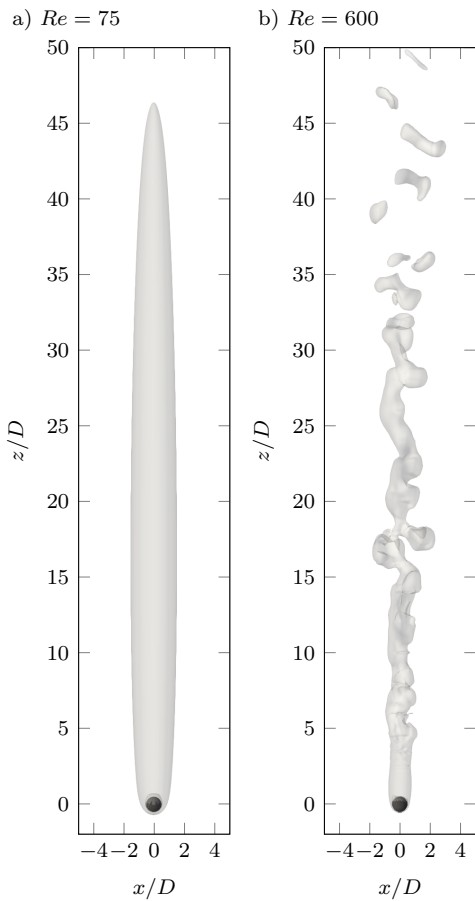

**Figure 4.** Isosurfaces of supersaturation in the wake at $T_\infty = -15°C$. The value of the isocontour is $s_{i,\infty} + 0.02$, i.e. two percentage points higher than the ambient supersaturation. Two different wake regimes are depicted, which correspond to two different hydrometeor sizes in our framework. (a) axisymmetric regime at $Re = 75$, (b) chaotic regime at $Re = 600$.

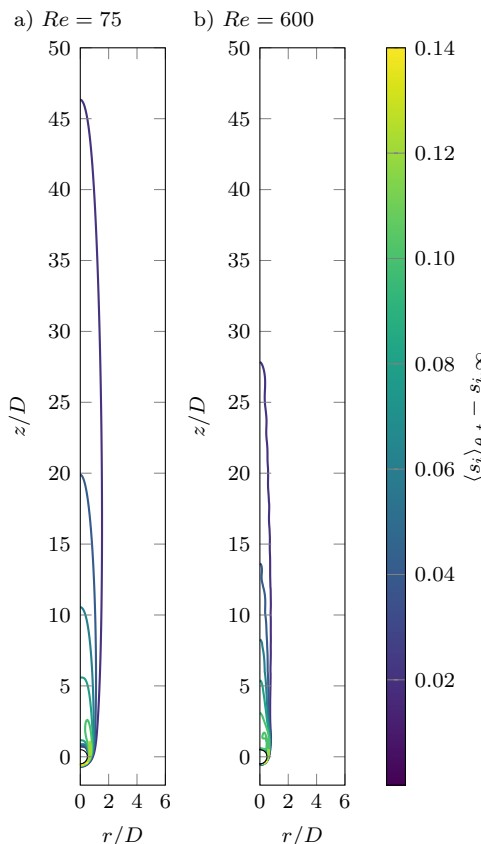

**Figure 5.** Contours of excess supersaturation in the wake, averaged over time and azimuthal direction at $T_\infty = -15°C$. (a) axisymmetric regime at $Re = 75$, (b) chaotic regime at $Re = 600$.

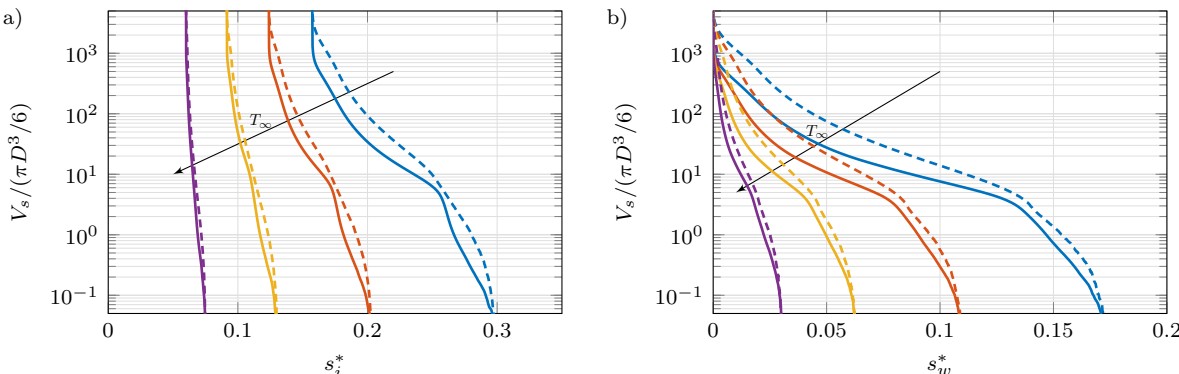

**Figure 6.** Volume of air where supersaturation exceeds a given threshold as a function of the threshold. The volume supersaturated w.r.t. to the solid phase is shown in (a), whereas (b) shows the volume supsaturated w.r.t. the liquid phase. The volume of the ice particle is used for normalization and four different ambient temperatures are shown: $T_\infty = -6°C$ (———), $T_\infty = -9°C$ (———), $T_\infty = -12°C$ (———). $T_\infty = -15°C$ (———). Solid lines correspond to $Re = 600$ (chaotic regime, time average), while dashed lines show the data obtained for $Re = 75$ (axisymmetric regime).

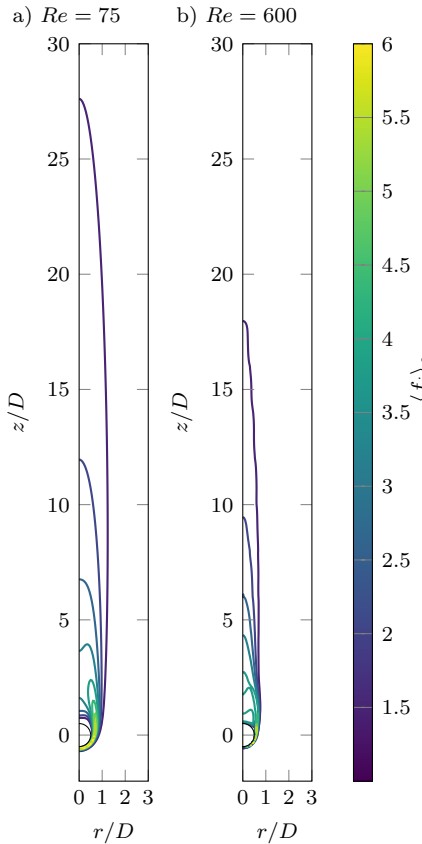

**Figure 7.** Contours of local ice enhancement factor in the wake, averaged over time and azimuthal direction at $T_\infty = -15°C$. (a) axisymmetric regime at $Re = 75$, (b) chaotic regime at $Re = 600$.

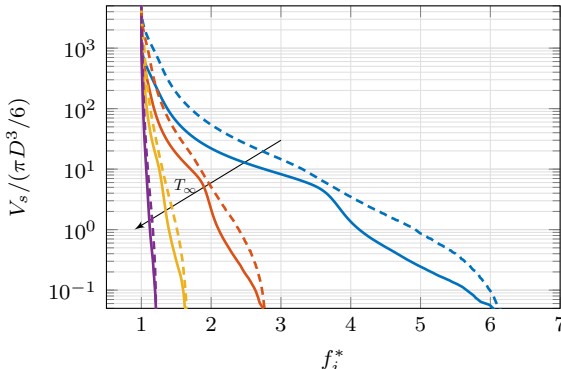

**Figure 8.** Volume of air with supersaturation above a given threshold as a function of the ice enhancement factor. The volume is normalized by the volume of the ice particle and four different temperatures are shown: $T_\infty = -6°C$ (──), $T_\infty = -9°C$ (──), $T_\infty = -12°C$ (──). $T_\infty = -15°C$ (──). Solid lines correspond to $Re = 600$ (chaotic regime, time average), while dashed lines show the data obtained for $Re = 75$ (axisymmetric regime).

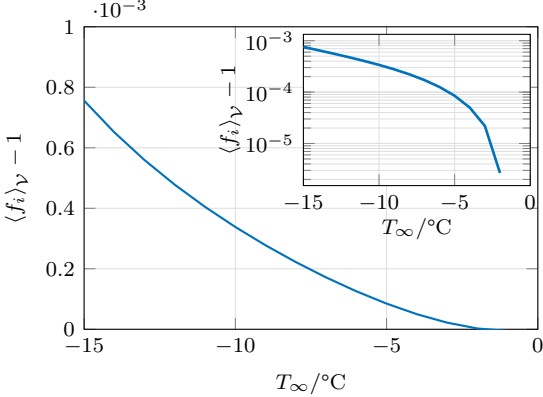

**Figure 9.** Global ice enhancement factor as a function of cloud temperature. The inset shows the same data, but in semi-logarithmic scale.

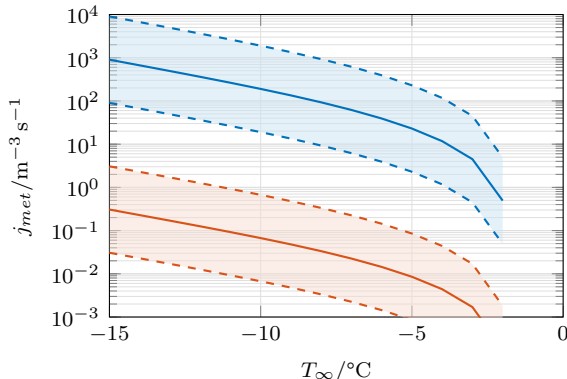

**Figure 10.** Limiting cases for the nucleation rate $j_{met}$. The swept-volume limited estimate based on the arguments of Prabhakaran et al. (2020) and evaluated from the present data according to Eq. (27) is shown for $C_{expo} = 10$ (——) with the shaded area depicting the values obtained for $1 < C_{expo} < 100$. The exposure-time limited estimate defined in Eq. (29), which is directly linked to the ice enhancement factor, is shown for $\tau_{nucl} = 10\text{s}$ (——) and the range $1\text{s} < \tau_{nucl} < 100\text{s}$ (shaded area).