# Peer review of "On the ice-nucleating potential of warm hydrometeors in mixed-phase clouds"

_Atmospheric Chemistry and Physics, 2020_

## Referee Comment (RC1) · Anonymous Referee #2 · 12 May 2020

Summary:

This study explores the possibility of the nucleation of ice crystals in the wake of hydrometeors undergoing wet growth through numerical simulation. The governing equations were non-dimensionalized using appropriate scales and solved using a spectral-element method. The control parameters are the Reynolds number and the temperature difference between the hydrometeor and the ambient. The spatially resolved simulation showed that the volume of the wake that is supersaturated is significantly greater than what was previously reported in the literature (Fukuta and Lee 1986). This is an interesting result. The authors transform the supersaturation in the wake to an ice nuclei concentration using a power law. Later, they average the ice nuclei concentration over the entire cloud volume and conclude that the increase in ice crystal

concentration is only marginal. The paper is fairly well written and organized. The volume analysis of the supersaturated wake is an important result and would be of interest to the mixed-phase community. However, the analysis on the ice enhancement in clouds is fairly weak. The main issues are the transformation of supersaturation to ice particle concentration using a power law, the choice of environmental parameters for wet growth and the estimation of the fractional cloud volume subjected to enhanced supersaturation due to the hydrometeors. This work should be considered for publication, but only after the comments listed below are addressed satisfactorily. The reviewer would like to revisit the manuscript after the comments are addressed. ================================

Major comments:

1) In the simulation, the surface temperature of the hydrometeor was fixed at 0 C. The ambient temperature was varied between -20 C and -40 C. Can the authors justify the choice of ambient temperatures for this study? Can the authors cite observations that detect wet growth at such low temperatures? There is a comprehensive experimental study by Greenan and List (JAS, 1995) on the surface temperature of hydrometeors at different conditions. It is unlikely that wet growth would occur at such low temperatures.

2) In section 3.3 the authors define a parameter called ice enhancement factor to quantify the effects of enhanced supersaturation. This parameter is justified, but the expression used for finding NIN is not. This expression is used in Baker 1991, but none of the recent work on ice nucleation use this expression (to the best of reviewer's knowledge). Such a power law relationship between the number concentration of ice nuclei and supersaturation seems physically inconsistent. For example, barring the effects of wettability/chemical composition, as the supersaturation is increased, the size of the aerosols that is activated is reduced. For ice nucleation, the size of the nucleus is an important parameter, and as the size of the nucleus is reduced, its ice nucleating efficiency is also reduced. So, the number concentration of ice nuclei may not increase with supersaturation like a power law with such high exponents as mentioned in this

paper. Furthermore, such a power law may not even be applicable to CCN concentrations when the supersaturation is quite high (Q. Ji and G. Shaw 1998 GRL). So, the applicability of such a power law to ice nuclei concentration is highly questionable. Can the authors comment/justify the applicability of the expression for NIN, as the whole of section 3.3 and the most important conclusion in the paper is based on this expression? This comment needs to be addressed in detail to support the conclusion. If this issue cannot be addressed satisfactorily, the authors can consider presenting their arguments based on fractional cloud volume (like in section 3.2) that is exposed to the enhanced supersaturation due to the falling wet hydrometeors.

3) The analysis in section 3.3 can be recast as the cloud volume that is exposed to very high supersaturation in the wake. This analysis concludes that the fraction of the cloud volume exposed to the high supersaturation in the wake is insignificant. There is a similar study published recently (Prabhakaran et al 2020 (GRL)). Their analysis concluded that a significant fraction of the cloud volume can be exposed to the high wake supersaturation during the lifetime of the cloud. Can the authors comment about the difference between these two analyses?

================================

Comments to improve the quality of the work:

1) In lines 108-109, the authors state that buoyancy contributions to momentum due to the variations in temperature and water vapor is negligible. Can the authors justify this statement briefly (a few lines) by quoting the value of the relevant parameter, e.g. Richardson number, along with the reference to Chouippe et al 2019? Would it be insignificant when the temperature difference between the ambient and the drop is 40 C? Similarly, in lines 118-119, can the authors justify briefly why the variations in the vertical velocity is not important in the present context?

2) In a deep convective cloud, the hydrometeors are falling through a turbulent environment. Can the authors comment about the role of turbulent fluctuations in the ambient?

How would the volume of the supersaturated region change with turbulence intensity in the ambient? There are some heat transfer studies from a heated sphere in a turbulent environment (Bagchi and Kottam 2008, Phys of Fluids). Can this be extended to the current study? It might be worthwhile to briefly discuss this as a part of future work.

3) Can the authors comment on how the supersaturated volume would be affected in the presence of cloud droplets and ice particles in the ambient?

==============================

Minor comment:

Excess supersaturation - notation difference between Eq. 15 and Fig 4 caption. Fig 4 caption has a "*" on top of "s".

---

## Referee Comment (RC2) · Alexei Korolev (Referee) · 18 Jun 2020

Review of the manuscript "On the ice-nucleating potential of warm hydrometeors in mixed-phase clouds" by Krayer et al.

**Overview:**

The paper presents the results of flow simulations of free falling particles focusing on calculations of high supersaturation formed in the particles' wake for the case when the environmental and particle temperatures are different. High supersaturation regions may result in the nucleation of INPs, which normally do not nucleate at low supersaturation. Under certain conditions, this process may result in the enhanced production of ice particles and explain one of the prevailing problems of secondary ice production. This study is a continuation of the work started in Chouippe et al. (2019). One of the important findings of this work is that regions with persistent high supersaturation, attained in the wake of warm hydrometeors, may extend up to 50 diameters downstream. The diagram in Fig.9 provides a clear assessment of the significance of this SIP mechanism at different temperatures. In my opinion, this is an important study further contributing in our understanding of the roles of different SIP mechanisms in ice initiation in clouds. The style, conciseness, and clarity of explanation left a very good impression. In my view, the paper should be published after addressing few comments listed below.

**Recommendation**:**

Accept after minor revision.

**Comments:**

- 1. Since it was not specified in the text, it appears that Eq.12 assumes that the lifetime of all hydrometeors is the same. This assumption would work well for riming particles falling through a mixed phase environment. However, the condition  $T_p=0^{\circ}$ C will be limited by the freezing time of drops and should be accounted for in Eq.12. Since freezing time for large and small drops may be different by few orders of magnitude (e.g. Murray and List, 1972), the effect of small droplets on the supersaturated volume may be lower than shown in Fig.6. The following results obtained in this work will also be affected. The effect of freezing time for the case of freezing drops requires clarification.
- 2. The supersaturation calculation was considered for a particle free falling in still air, i.e. in a nonturbulent environment ( $\varepsilon = 0$ ). Could you speculate on a qualitative level, how  $\varepsilon > 0$  may affect your results? Would it increase or decrease the global ice enhancement factor?
- 3. What is the role of air pressure *P* on the results obtained in this paper (specifically Fig.9)? Since particle fall speed, viscosity and thermal conductivity depend on *P*, it may have a noticeable effect on the mixing rate, supersaturated volume, amplitude of result supersaturation and the persistence of supersaturated regions. It is worth indicating what *P* was used in this study.
- 4. An important element not discussed in this study is the conceptual consideration of how this SIP process is related to natural clouds and identification of environmental conditions when it becomes significant. There are a few statements regarding this matter scattered throughout the manuscript. However, it leaves the reader with an impression of incompleteness of this paper. For example, as discussed in section 2, the condition  $T_p=0^{\circ}$ C can be satisfied for riming ice particle or freezing drops. For the first case, the accretion of cloud droplets on the ice surface should reach the wet growth regime, i.e. when LWC reach the Ludlam limit. At -30C, for a free-falling hailstone, the Ludlam limit exceeds 5g/m3 (the exact number needs to be checked). Such high LWC at -30C

does not seem to be feasible. Regarding the second case, there are very few reports on observations of precipitation size drops (D>100um) at -30C. Therefore, this option also appears to be uncommon in clouds. In addition to the discussion on page 15 related to Fig.9, it is worth expanding the discussion about the feasibility and significance of this SIP mechanism in temperatures warmer than -30C. Mentioning that convective clouds are the most likely candidates for this type of SIP to occur would be also relevant.

- 5. Eq.10: What Dj,min and Dj,max were used in this study? It is worth indicating the in the text.
- 6. Definition of  $\tilde{s}_i^*$  is worth introducing in the text prior to Fig.4.
- 7. Line 201: Should it be  $\tilde{s}_i^*$ ?
- 8. Eq.12. Definitions of  $\tau$  and  $\Omega$  should be provided in the text.
- 9. Line 206: "... as a function of the threshold". I guess you employed  $\tilde{s}_i^*=0.1$  threshold. This should be indicated in the text.
- 10. Lines 24 and 358: Field et al. 2016 should be 2017.

Alexei Korolev

---

## Author Comment (AC1) · 15 Sep 2020

**Responses to reviewer 1: discussion (acp-2020-136)**

September 15, 2020

The authors would like to thank Alexei Korolev for taking the time to review the manuscript and for his constructive feedback. We hope the responses provided below help to clarify the issues addressed. The adjustments made to the manuscript resulting from the feedback are summarized at the end of each response.

**Remark by the authors** In the original "referee comment", the reviewer is not referring to the latest version of the manuscript, but to the pre-discussion version. The line numbers and equation numbers have been adapted in the following to reflect the discussion paper.

**1 Comments**

R1-Co1. Since it was not specified in the text, it appears that Eq. 16 assumes that the lifetime of all hydrometeors is the same. This assumption would work well for riming particles falling through a mixed phase environment. However, the condition  $T_p = 0^{\circ}$ C will be limited by the freezing time of drops and should be accounted for in Eq. 16. Since freezing time for large and small drops may be different by few orders of magnitude (e.g. Murray and List, 1972), the effect of small droplets on the supersaturated volume may be lower than shown in Fig. 6. The following results obtained in this work will also be affected. The effect of freezing time for the case of freezing drops requires clarification.

The results of this work should be considered as a best-case scenario for ice enhancement, i.e. how much ice enhancement is to be expected at very favorable conditions. We agree with the reviewer that the assumption of  $T_p = 0$ °C is more suitable for riming ice than for freezing drops, and it will be stated more clearly in the revised manuscript that riming ice is the main mechanism considered in this work.

When freezing drops are considered,  $T_p = 0^{\circ}$ C is still a good approximation for the surface temperature in case of inward freezing (Johnson and Hallett, 1968), however, unlike in the case of riming ice, there is no mechanism which sustains the gap in surface to ambient temperature after the freezing time is exceeded. The temperature distribution (for a given meteor size) will therefore not be concentrated on a single value, and instead, the rate at which new unfrozen droplets are available, the rate at which fully frozen droplets deplete and the size population dynamics with history effects have to be taken into account. Qualitatively, ice enhancement will be lower than the estimation given in the present work when the freezing time is taken into consideration. Because the freezing time is correlated positively to the meteor size, this deviation is expected to be more significant at low cloud temperatures, since the size distribution shifts towards lower values the lower the cloud temperature is. Therefore, freezing drops are even less likely to generate significant amounts of secondary ice than riming ice particles.

Adjustments to the manuscript: A paragraph will be added to the manuscript to emphasize this point.

R1-Co2. The supersaturation calculation was considered for a particle free falling in still air, i.e. in a non- turbulent environment ( $\epsilon = 0$ ). Could you speculate on a qualitative level, how  $\epsilon > 0$  may affect your results? Would it increase or decrease the global ice

**enhancement factor?**

The study of Bagchi and Kottam (2008) is very helpful when the effect of ambient turbulence on the supersaturated volume and ice enhancement is discussed. The parameter range investigated in their work corresponds reasonably well to the scenario of a hydrometeor with a diameter of a few millimeter settling under atmospheric conditions, where the largest flow scales are expected to be  $\mathcal{O}(100\text{m})$  and the smallest scales around  $\mathcal{O}(1\text{mm})$  (Lehmann, Siebert, and Shaw, 2009).

We know from simple mixing parcel models (e.g. (Chouippe, Krayer, Uhlmann, Dušek, Kiselev, and Leisner, 2019, fig. 17) or (Prabhakaran, Kinney, Cantrell, Shaw, and Bodenschatz, 2020, fig. 4)) that the highest supersaturations occur in regions where the temperature of the mixture is roughly halfway between  $T_{\infty}$  and  $T_p$ , i.e.  $\tilde{T} \approx 0.5$  with  $\tilde{T}$  being the non-dimensionalized temperature as introduced in the manuscript. In (Bagchi and Kottam, 2008, fig. 17) it is demonstrated that the centerline temperature in the wake decays significantly faster if the background flow is turbulent, especially when  $\tilde{T} \leq 0.4$ . Therefore it is to be expected that supersaturation decays faster in the wake than it does for a uniform inflow, and thus, the supersaturated volume as well as the ice enhancement are most likely smaller if the ambient is turbulent.

However, Bagchi and Kottam (2008) also investigated the the effect of turbulence on the heat and mass transfer coefficient. While the mean value of the Nusselt number remains mostly unaffected, strong fluctuations in its value can be observed. This presumably leads to a more intermittent behavior of the temperature and vapor fields. The role of intermittency on ice nucleation activity still needs to be investigated more thoroughly, especially when the distribution of aerosol particles is explicitly considered (a point which was suggested to be investigated as part of future work). If regions of strong supersaturation conincide with regions where AP are preferentially located, intermittency might promote ice nucleation as supersaturation and nucleation rate are non-linearly linked to the temperature/vapor fields.

Adjustments to the manuscript: A paragraph discussing the influence of turbulence will be added to the manuscript.

*Note:* This response is the same as the response to remark R2-Co2 raised by reviewer 2, due to the strong similarity of the remarks.

R1-Co3. What is the role of air pressure P on the results obtained in this paper (specifically Fig. 9)? Since particle fall speed, viscosity and thermal conductivity depend on P, it may have a noticeable effect on the mixing rate, supersaturated volume, amplitude of result supersaturation and the persistence of supersaturated regions. It is worth indicating what P was used in this study.

For the numerical similations, no assumptions on the actual value of pressure have to be made since the incompressible Navier-Stokes equations are independent of the absolute value of pressure and the only imposed parameters in our framework are the particle Reynolds number, the Prandtl number and the Schmidt number, whose definitions are

$$Re = \frac{v_p D}{\nu}, \ Pr = \frac{\nu}{D_T}, \ Sc = \frac{\nu}{D_{n_v}}$$

where the dependency on the fluid density is approximately  $\nu \sim \rho^{-1}$ ,  $D_T \sim \rho^{-1}$  and  $D_{n_v} \sim \rho^{-1}$  in the low density limit (see e.g. Bird, Stewart, and Lightfoot (2002)). Throughout this work, air is assumed to be an ideal gas and therefore

$$P \propto \rho T.$$
 (1)

The only parameter affected by a variation in pressure is therefore the Reynolds number due to the modification in viscosity and fall speed of the hydrometeor. However, in the majority of the manuscript the dependency on the Reynolds number is neglected except for transitions in flow regimes. Therefore, the only relevant quantity within the framework of this work which is affected by the absolute value of pressure is the threshold diameter for regime transition (see response to R1-Co5 for the relationship). However, this quantity is already subject to various uncertainties such as the shape of the hydrometeor (affecting fall speed and flow regimes) and its density such that taking into account different pressure levels does not seem worthwhile. The supersaturated volume scaled by the hydrometeor volume is of

similar order of magnitude for both regimes investigated. Therefore, a shift in threshold diameter for transition alone is unable to lead to a significant alteration of the global ice enhancement factor.

In this work, the density of ambient air was assumed to be  $1 \text{ kg m}^{-3}$  which approximately corresponds to the values of atomospheric pressure presented in table 1 in the temperature range investigated according to the ideal gas equation of state.

| temperature [°C] | -40 | -30 | -20 | -10 | 0   |
|------------------|-----|-----|-----|-----|-----|
| pressure [kPa]   | 669 | 698 | 727 | 755 | 783 |

Table 1: Ambient air pressure for different air temperatures investigated.

R1-Co4. An important element not discussed in this study is the conceptual consideration of how this SIP process is related to natural clouds and identification of environmental conditions when it becomes significant. There are a few statements regarding this matter scattered throughout the manuscript. However, it leaves the reader with an impression of incompleteness of this paper. For example, as discussed in section 2, the condition  $T_p = 0^{\circ}$ C can be satisfied for riming ice particle or freezing drops. For the first case, the accretion of cloud droplets on the ice surface should reach the wet growth regime, i.e. when LWC reach the Ludlam limit. At  $-30^{\circ}$ C, for a free-falling hailstone, the Ludlam limit exceeds  $5 \text{g m}^{-3}$  (the exact number needs to be checked). Such high LWC at  $-30^{\circ}$ C does not seem to be feasible. Regarding the second case, there are very few reports on observations of precipitation size drops ( $D > 100 \mu m$ ) at  $-30^{\circ}C$ . Therefore, this option also appears to be uncommon in clouds. In addition to the discussion on page 11 related to Fig. 9, it is worth expanding the discussion about the feasibility and significance of this SIP mechanism in temperatures warmer than  $-30^{\circ}$ C. Mentioning that convective clouds are the most likely candidates for this type of SIP to occur would be also relevant.

The reviewer is correct that wet-growth at low cloud temperatures only occurs if the LWC is sufficiently high, and we arrive at a similar estimation for the Ludlam limit when applying the empirical formula of García-García and List (1992),

$$W_{f,SLL} = -2 \cdot 10^{-4} \frac{\text{kg}}{\text{m}^3 \,^{\circ}\text{C}} T_{\infty},$$
 (2)

which results in  $W_{f,SLL} = 6 \text{g m}^{-3}$  for  $T_{\infty} = -30^{\circ}\text{C}$ . Such high LWC and temperature gaps between the hydrometeors surface and the ambient exceed the range of values commonly observed in natural clouds, and merely served the purpose of demonstrating that exceptional/unrealistic conditions are required for this mechanism to be relevant. In accordance to the response given in R2-Ma1, we have decided to adjust the temperature range investigated and to focus on temperatures warmer than  $-15^{\circ}\text{C}$ . The discussion on fig. 9 will be rewritten accordingly. The revised versions of all affected figures may be found in section 2 of this document.

We will also add a paragraph discussing the required cloud conditions to the manuscript, where the relevance of high LWC is emphasized. We agree that convective clouds are the most likely candidates for the required conditions.

Adjustments to the manuscript: The temperature range investigated is adjusted. Revised versions of all affected figures can be found in section 2 of this document. The text in the manuscript will be adjusted accordingly. A paragraph discussing the required environmental conditions and possible cloud types in which they occur is added to the manuscript.

**R1-Co5.** Eq.10: What $D_{j,min}$ and $D_{j,max}$ were used in this study? It is worth indicating the in the text.**

The diameter thresholds were determined from the critical Reynolds numbers of regime transition  $Re_c$

using the following set of equations.

$$Re_{c,j} = D_{j,max} |v_p|/\nu$$
$$v_p = u_g \left(3C_d(Re_{c,j})/4\right)^{-1/2}$$
$$u_g = \sqrt{\left(\rho_p/\rho_f - 1\right)|g|D_{j,max}}$$

The drag coefficient is approximated using the empirical drag law of Schiller and Naumann (1933). The density of the hailstone is assumed to be  $600 \text{kg m}^{-3}$ , as has been stated in the main text, and the fluid density was set to  $1 \text{kg m}^{-3}$  (see also the response to R1-Co3 concerning this). Furthermore, the kinematic viscosity was set constant to  $1 \cdot 10^{-5} \text{m}^2 \text{s}^{-1}$  in the discussion paper, but has been revised to a value of  $1.68 \cdot 10^{-5} \text{m}^2 \text{s}^{-1}$  in the revised version of the manuscript, which corresponds approximately to the value at 750 kPa air pressure and  $-10^{\circ}\text{C}$  ambient temperature.

For Fig. 2 and Fig. 3 in the discussion paper, all four regimes are considered and the values of  $Re_c$  have been chosen in accordance to the values stated in section 3.1 of the discussion paper. The threshold diameters used in the discussion paper  $(D_{min}^{old}, D_{max}^{old})$  and the revised manuscript  $(D_{min}^{new}, D_{max}^{new})$  are summarized in table 2.

| regime              | $D_{min}^{old}$   | $D_{max}^{old}$   | $D_{min}^{new}$   | $D_{max}^{new}$   | $Re_c$ |
|---------------------|-------------------|-------------------|-------------------|-------------------|--------|
| axisymmetric        | 0                 | $0.77\mathrm{mm}$ | 0                 | $1.08\mathrm{mm}$ | 212    |
| steady oblique      | $0.77\mathrm{mm}$ | $0.88\mathrm{mm}$ | $1.08\mathrm{mm}$ | $1.24\mathrm{mm}$ | 273    |
| oscillating oblique | $0.88\mathrm{mm}$ | $1.0\mathrm{mm}$  | $1.24\mathrm{mm}$ | $1.44\mathrm{mm}$ | 360    |
| chaotic             | $1.0\mathrm{mm}$  | $\infty$          | 1.44 mm           | $\infty$          | _      |

Table 2: Threshold diameters for regime transition used in the discussion paper  $(D_{min}^{old}, D_{max}^{old})$  and the revised manuscript  $(D_{min}^{new}, D_{max}^{new})$ .

From section 3.2 onward, only the axisymmetric and chaotic regimes are considered and the following the shold diameter has been chosen for concreteness.

| regime       | $D_{min}^{old}$   | $D_{max}^{old}$   | $D_{min}^{new}$   | $D_{max}^{new}$   |
|--------------|-------------------|-------------------|-------------------|-------------------|
| axisymmetric | 0                 | $0.85\mathrm{mm}$ | 0                 | $1.26\mathrm{mm}$ |
| chaotic      | $0.85\mathrm{mm}$ | $\infty$          | $1.26\mathrm{mm}$ | $\infty$          |

Adjustments to the manuscript: The threshold diameters will be stated in the text.

**R1-Co6.** Definition of $\tilde{s}_i^*$ is worth introducing in the text prior to Fig.4.**

We agree that either the caption should be rephrased or the concept  $\tilde{s}_i$  should be introduced in the text before referring to the figure.

**R1-Co7.** Line 222: Should it be $\tilde{s}_i^*$ ?**

In this case, we are not referring to a threshold (as we do for the supersaturated volume), but to the supersaturation field itself, i.e.  $\tilde{s}_i = \tilde{s}_i(\mathbf{x}, t)$ . The expression in line 222 may be rephrased as  $s_i(\mathbf{x}, t) > s_{i,\infty}$ .

**R1-Co8.** Eq.16. Definitions of $\tau$ and $\Omega$ should be provided in the text.**

We thank the reviewer for bringing this to our attention. Indeed the definitions of  $\tau$  and  $\Omega$  are absent in the text and will be added in the revised version of the manuscript.  $\tau$  denotes the time over which the flow is time-averaged and  $\Omega$  denotes the simulation domain, i.e. in this case a volume integral is computed over the entire domain of observation.

**R1-Co9. Line 227: "... as a function of the threshold". I guess you employed $\tilde{s}_i^* = 0.1$ threshold. This should be indicated in the text.**

This appears to be a misunderstanding. Eq. (16) in the discussion paper is an expression which

computes the volume of fluid in the domain of observation which is supersaturated at a value higher than a given value (the threshold  $s_i^*$ ), i.e. it answers the question "How much fluid volume exhibits a supersaturation of at least  $s_i^*$ ?". In figure 6 of the discussion paper, the resulting volume is shown for varying thresholds, which are in the figure given as offsets from  $s_{i,\infty}$ . However, for reasons explained in the response to R2-Ma2, the supersaturated volume will be given as a function of  $s_i^*$  instead of  $\tilde{s}_i^* = s_i^* - s_{i,\infty}$  in the revised version of the manuscript (see fig. 3 of this document).

**R1-Co10. Lines 24 and 358: Field et al. 2016 should be 2017.**

We thank the reviewer for pointing out this mistake, which will be corrected in the revised version of the manuscript.

**2 Figures**

Figure 1: Isosurfaces of supersaturation in the wake at  $T_{\infty} = -15^{\circ}$ C. The value of the isocontour is  $\tilde{s}_{i}^{*} = 0.02$ , i.e. two percentage points higher than the ambient supersaturation. Two different wake regimes are depicted, which correspond to two different hydrometeor sizes in our framework. (a) axisymmetric regime at Re = 75, (b) chaotic regime at Re = 600.

Changelog: changed ambient temperature to  $T_{\infty}=-15^{\circ}{\rm C}$ ; changed isocontour threshold to  $\tilde{s}^*_i=0.02$ ; adapted caption accordingly

Figure 2: Contours of excess supersaturation in the wake, averaged over time and azimuthal direction at  $T_{\infty} = -15^{\circ}$ C. (a) axisymmetric regime at Re = 75, (b) chaotic regime at Re = 600. Changelog: changed ambient temperature to  $T_{\infty} = -15^{\circ}$ C; adapted caption accordingly

Figure 3: Volume of air where supersaturation w.r.t. ice exceeds a given threshold as a function of the threshold. The volume is normalized by the volume of the ice particle and four different ambient temperatures are shown:  $T_{\infty} = -6^{\circ}C$  (----),  $T_{\infty} = -9^{\circ}C$  (-----),  $T_{\infty} = -12^{\circ}C$  (-----).  $T_{\infty} = -15^{\circ}C$  (-----). Solid lines correspond to Re = 600 (chaotic regime), while dashed lines show the data obtained for Re = 75 (axisymmetric regime).

Changelog: changed ambient temperature range; added  $T_\infty$  indicator; adapted caption accordingly

---

## Author Comment (AC2) · 15 Sep 2020

**Responses to reviewer 2: discussion (`acp-2020-136`)**

September 15, 2020

The authors would like to thank the anonymous referee for taking the time to review the manuscript and for addressing various important issues which helped us improving the quality of our work. We have decided to implement substantial changes to our manuscript based on his remarks, the details of which are addressed point by point below.

**1    Major remarks**

**R2-Ma1. In the simulation, the surface temperature of the hydrometeor was fixed at $0°$C. The ambient temperature was varied between $-20°$C and $-40°$C. Can the authors justify the choice of ambient temperatures for this study? Can the authors cite observations that detect wet growth at such low temperatures? There is a comprehensive experimental study by Greenan and List (JAS, 1995) on the surface temperature of hydrometeors at different conditions. It is unlikely that wet growth would occur at such low temperatures.**

We agree with the reviewer that the temperature gap investigated in this work exceeds the range of values which are to be reasonably expected for natural clouds. The intention behind this was to demonstrate that extreme conditions are required in order to produce significant wake-induced ice enhancement, and that these conditions are unlikely to be observed in nature. However, this point apparently has not been communicated clearly enough. Furthermore, the decision to portray the contour plots at $T_\infty = -30°$C seems unfortunate. We have therefore decided to vary the ambient temperature in a smaller range ($-15°$C $< T_\infty < 0°$C) according to the experimental observations by Greenan and List (1995) and depict the contours at $T_\infty = -15°$C.
*Adjustments to the manuscript:* The adjusted versions of the affected figures can be found in section 4 of this document. The text in the manuscript will be adjusted accordingly.

**R2-Ma2. In section 3.3 the authors define a parameter called ice enhancement factor to quantify the effects of enhanced supersaturation. This parameter is justified, but the expression used for finding NIN is not. This expression is used in Baker 1991, but none of the recent work on ice nucleation use this expression (to the best of reviewer's knowledge). Such a power law relationship between the number concentration of ice nuclei and supersaturation seems physically inconsistent. For example, barring the effects of wettability/chemical composition, as the supersaturation is increased, the size of the aerosols that is activated is reduced. For ice nucleation, the size of the nucleus is an important parameter, and as the size of the nucleus is reduced, its ice nucleating efficiency is also reduced. So, the number concentration of ice nuclei may not increase with supersaturation like a power law with such high exponents as mentioned in this paper. Furthermore, such a power law may not even be applicable to CCN concentrations when the supersaturation is quite high (Q. Ji and G. Shaw 1998 GRL). So, the applicability of such a power law to ice nuclei concentration is highly questionable. Can the authors comment/justify the applicability of the expression for NIN, as the whole of section 3.3 and the most important conclusion in the paper is based on this expression? This comment needs to be addressed in detail to support the conclusion. If this issue**

**cannot be addressed satisfactorily, the authors can consider presenting their arguments based on fractional cloud volume (like in section 3.2) that is exposed to the enhanced supersaturation due to the falling wet hydrometeors.**

The power-law equation for $N_{IN}$ indeed appears to be rarely used in recent literature. We have therefore decided to replace it by the exponential law provided by Meyers, DeMott, and Cotton (1992) which has been obtained from continuous-flow diffusion chamber (CFDC) measurements of natural aerosols. The constitutive relation is a parametrization of both the deposition and condensation-freezing mechanisms of ice nucleation and reads

$$N_{IN} = 0.528 \exp\left(12.96 s_i\right) \mathrm{m}^{-3}. \tag{1}$$

It is reported to be strictly valid for the following parameter range (the range of the CFDC data).

$$-20°\mathrm{C} < T < -7°\mathrm{C}, \quad 2\% < s_i < 25\%, \quad -5\% < s_w < 4.5\% \tag{2}$$

The temperatures of interest in the current work (after making the adjustments stated in R2-Ma1) fall well into the range of validity. The distribution of $s_i$ in the wake is shown in fig. 3 (an updated version of fig. 6 of the discussion paper). For most ambient temperatures, $s_i$ does not exceed 25%. At $T_\infty = -15°\mathrm{C}$, regions where $s_i$ is slightly larger than 0.25 exist, but only occupy a small volume within the domain, and hence imprecisions are likely to be insignificant for integral quantities. When looking at the volumetric distribution of $s_w$ in the wake in fig. 4, it can be seen that water supersaturation exceeds the CFDC data range in significant portions of the domain when $T_\infty \lesssim -10°\mathrm{C}$. Due to these relatively large supersaturations w.r.t. liquid, eq. (1) might underestimate the contribution of the condensation-freezing mode, as this mode shows increased activity under these conditions as has been demonstrated by Schaller and Fukuta (1979) for various substances. However, we are not aware of any parametrization of condensation-freezing nucleation for natural aerosols which can be directly applied under these conditions.

The main conclusion of this work is not affected by the substition of the nucleation law. The global ice enhancement factor computed with eq. (1) behaves similar to the power law estimation with $\alpha \approx 3$ as can be seen when comparing fig. 7 of this document to fig. 9 of the discussion paper.

*Adjustments to the manuscript:* All affected figures have been updated and the text will be adjusted accordingly. The range of validity will be justified similar to the response above. Figure 4 will be added to the manuscript as it raises awareness concerning the applicability of eq. (1) and provides information on the supersaturation w.r.t. liquid which may be interesting for the reader.

**R2-Ma3. The analysis in section 3.3 can be recast as the cloud volume that is exposed to very high supersaturation in the wake. This analysis concludes that the fraction of the cloud volume exposed to the high supersaturation in the wake is insignificant. There is a similar study published recently (Prabhakaran et al 2020 (GRL)). Their analysis concluded that a significant fraction of the cloud volume can be exposed to the high wake supersaturation during the lifetime of the cloud. Can the authors comment about the difference between these two analyses?**

It is true that the current manuscript investigates the instanteneous exposure of a cloud subvolume to meteor-induced supersaturation, while the analysis presented in Prabhakaran, Kinney, Cantrell, Shaw, and Bodenschatz (2020) focuses on the volume swept by the meteors. The latter approach is reasonable since history effects in ice nucleation should be taken into account, i.e. it should be taken into account that ice nuclei which have been activated in the wake of a hydrometeor may stay activated once they are not exposed to the wake anymore. However, the difference in the two analyses can be regarded as two limiting cases of the nucleation rate, namely one which is limited by the rate of renewal of fluid in the wake (analysis of Prabhakaran et al. (2020)) and one which is limited by the time scale of nucleation (our analysis), as will be demonstrated in the following.

Following Prabhakaran et al. (2020), the time required for a significant volume of air to be sampled by hydrometeor wakes is estimated by

$$\tau_{sweep} = \left(\int_0^\infty \bar{N}_{met}(D)\dot{V}_{sweep}(D)\,\mathrm{d}D\right)^{-1}, \tag{3}$$

where $\bar{N}_{met}(D)$ denotes the number concentration density of ice particles and $\dot{V}_{sweep} = \epsilon v_p D^2 \pi/4$ is the volumetric flow rate of air which is swept by a hydrometeor with diameter $D$ and velocity $v_p$. The unknown factor $\epsilon$ is assumed to be of the order of unity. Using eq. (10) of the discussion paper for $\bar{N}_{met}(D)$ and the terminal velocity for smooth spheres, we obtain $\tau_{sweep} \approx 110$s at $T = -15°$C, which fits the estimation of Prabhakaran et al. (2020) well. The use of an empirical law for the terminal velocity of frozen hydrometeors of natural shape leads to longer time scales, however, they are found to be of similar order of magnitude. This analysis tells us that even though the cloud volume which is instanteneously exposed to high supersaturations is very small, it does not take a long time to expose a significant volume because the rate at which air is swept by the meteors is high.

In the following we attempt to quantify the nucleation rate of INP, henceforth denoted as $j_{met}$, from our simulation data and the swept-volume argument. Under the assumption that nucleation occurs sufficiently fast to achieve the INP concentrations predicted by eq. (1), the nucleation rate can be estimated from the number of INP activated in the wake and the time it takes to replenish the volume of fluid affected by high supersaturations. The former is obtained directly from our simulation data by computing the volume integral $\int_{\Omega(D)} \left( N_{IN}(\mathbf{x}) - N_{IN,\infty} \right) d\mathbf{x}$ while the latter is difficult to define objectively. We propose to estimate the time scale of wake renewal by

$$\tau_{expo} = \frac{V_{aff}}{\dot{V}_{sweep}} \tag{4}$$

where $V_{aff} = \gamma D^3 \pi/6$ is the volume affected by the wake of a hydrometeor of diameter $D$, which should be proportional to the volume of the hydrometeor. The prefactor $\gamma$ is currently unknown, but might be related to the concept of supersaturated volume defined in the manuscript. The time scale $\tau_{expo}$ may be regarded as the characteristic time a fluid volume is exposed to high supersaturations, and hence the subscript. From the definitions of $V_{aff}$ and $\dot{V}_{sweep}$ it follows that

$$\tau_{expo} \propto \frac{D}{v_p}, \tag{5}$$

with the constant of proportionality being referred to as $C_{expo}$ hereafter. This new constant contains both unknown coefficients $\epsilon$ and $\gamma$ and might be interpreted as the non-dimensional streamwise length of the wake. Again, this length is difficult to define rigorously due to the asymptotic decay of supersaturation. However, judging from fig. 5 it is likely that $C_{expo} = \mathcal{O}(10)$ which results in exposure times of the order of $\tau_{expo} \approx 5$ms for all diameters of interest. The swept-volume limited nucleation rate for an ensemble of meteors is then given by

$$\dot{j}_{met}^{expo} = \int_0^\infty \bar{N}_{met}(D) \frac{1}{\tau_{expo}(D)} \int_{\Omega(D)} \left( N_{IN}(\mathbf{x}) - N_{IN,\infty} \right) d\mathbf{x} \, dD \tag{6}$$

under the assumptions that the rate of activation of INP is sufficiently fast, a sufficient number of interstitial aerosol particles are present and that those are homogeneously distributed within the wake.

The exposure time estimated previously is substantially shorter than the time scales usually relevant for cloud modelling (few milliseconds compared to minutes). As eq. (1) has been developed for cloud modelling, the validity of the assumption that INP activation can be regarded as instantaneous at the time scales considered should be brought into question. Indeed, classical nucleation theory (Fletcher, 1958) suggests that nucleation is a time-dependent process until the activated fraction of AP approaches unity. From concentrations of INP obtained from continuous-flow diffusion chamber (CFDC) experiments, the nucleation rate may be estimated by taking into account the residence time in the apparatus $\tau_{nucl}$ (Hoose and Möhler, 2012). Since eq. (1) is based on CFDC data, we make the conjecture that the local nucleation rate may be approximated by the relationship

$$j_{IN}(\mathbf{x}) \approx N_{IN}(\mathbf{x})/\tau_{nucl}. \tag{7}$$

In Hoose and Möhler (2012) residence times ranging from 1.6s to 120s are reported for various CFDC experiments. The primary data used to obtain eq. (1) also suggests that the peak concentration $N_{IN}$ is achieved with residence times of approximately 10s (Al-Naimi and Saunders, 1985, fig. 6) and that

shorter residence times lead to lower concentrations (in accordance to the arguments stated before). As can already be seen, $\tau_{nucl} \gg \tau_{expo}$, and hence, the supposition that the INP concentrations predicted by eq. (1) are achieved within the exposure time is disproved. The rate-limited nucleation rate of the ensemble of hydrometeors is then given by

$$j_{met}^{nucl} = \int_0^\infty \bar{N}_{met}(D) \frac{1}{\tau_{nucl}} \int_{\Omega(D)} \left( N_{IN}(\mathbf{x}) - N_{IN,\infty} \right) d\mathbf{x} \, dD, \tag{8}$$

under the assumption that interstitial AP are entrained sufficiently fast into the wake (which is reasonable given the arguments by Prabhakaran et al. (2020)) and that they are distributed homogeneously within the wake.

Figure 8 shows the meteor-induced nucleation rate for both limiting cases. We assume that the most likely values for the tunable parameters are $C_{expo} = 10$ and $\tau_{nucl} = 10$s (solid lines), but also investigate the range $C_{expo} \in [1, 100]$ and $\tau_{nucl} \in [1, 100]$ s in order to pay regard to the uncertainties associated with these quantities (shaded area). The swept-volume limited estimation $j_{met}^{expo}$ is at least two orders of magnitude higher than rate-limited estimation $j_{met}^{nucl}$. A high relevance of the wake-induced nucleation is indicated by $j_{met}^{expo}$, as it would only take around 40s for the number concentration of wake-activated INP to match the concentration of primary meteors at $T_\infty = -10$°C. In contrast, it would take around 82h to achieve this concentration with the rate-limited estimation, which suggests that this process is of little relevance in clouds. The large disparity between the results is explained by the differences in time scales, i.e. $\tau_{expo} = \mathcal{O}(10^{-3}\text{s})$ while $\tau_{nucl} = \mathcal{O}(10^1\text{s})$, as has been stated earlier. Physically this implies that the time a fluid volume is exposed to high supersaturations is too short to create considerable concentrations of INP.

This result can be linked to the ice enhancement factor introduced in the manuscript, as this quantity directly relates to the rate-limited estimation of the nucleation rate:

$$\langle f_i \rangle_{\mathcal{V}} = \frac{j_{met}^{nucl}}{N_{IN,\infty}/\tau_{nucl}} + 1. \tag{9}$$

Furthermore, it is straightforward to show that $\tau_{expo} \propto \tau_{sweep}$ for a given meteor concentration, which implies that as soon as a significant cloud volume is swept quickly at low volume fractions of ice, the transient exposure of a cloud fluid element to the wake will be short.

In order for wake-induced ice nucleation to be a relevant SIP, the nucleation rate in the wake needs to be significantly higher than what has been estimated in this work. If the conjectures presented in the above analysis hold, the most feasible way to accomplish this is that the overall concentration of AP is higher than what has been assumed in this work implicitly through eq. (1), i.e. this mechanism may gain importance in clouds with a high number of possible nucleation sites. It might also be conceivable that the AP concentration is locally enhanced in the wake due to flow-induced clustering, i.e. AP may be preferentially located in highly supersaturated region as opposed to the ambient, although this mechanism unlikely leads to the required concentrations. Nonetheless, we suggest that an analysis of individual AP trajectories may be beneficial in the future to clarify the importance of this SIP, as such an analysis would allow for a more rigorous assessment of the nucleation rate by providing access to the actual residence times of AP and by enabling the use of more fundamental constitutive laws for ice nucleation (such as classical nucleation theory).

*Adjustments to the manuscript:* Subsection 3.3 will be fully revised and extended by the discussion above. Figure 8 will be added to the manuscript. The notation concerning ice nuclei concentration and the concentration density of primary hydrometeors will be adjusted.

**2   Comments**

**R2-Co1. In lines 108-109, the authors state that buoyancy contributions to momentum due to the variations in temperature and water vapor is negligible. Can the authors justify this statement briefly (a few lines) by quoting the value of the relevant parameter, e.g. Richardson number, along with the reference to Chouippe et al 2019? Would it be insignificant when the temperature difference between the ambient and the drop is $40$°C?**

**Similarly, in lines 118-119, can the authors justify briefly why the variations in the vertical velocity is not important in the present context?**

The Richardson number for a freely falling heated sphere has been defined in Chouippe, Krayer, Uhlmann, Dušek, Kiselev, and Leisner (2019) as

$$Ri_T = \frac{1}{\left(\frac{\rho_p}{\rho_\infty} - 1\right)} \frac{T_p - T_\infty}{T_\infty}, \tag{10}$$

where $T_\infty$ is given in Kelvin (see (Chouippe et al., 2019, Appendix A) for the derivation). In accordance to the value stated in the discussion paper, we assume $\rho_p = 600 \text{kg m}^{-3}$ and $\rho_\infty = 1 \text{kg m}^{-3}$. For an ambient temperature of $T_\infty = -40°C = 233.15K$ and a particle temperature of $T_\infty = 0°C = 273.15K$, we obtain

$$Ri_T \approx 3 \cdot 10^{-4}. \tag{11}$$

In (Chouippe et al., 2019, fig. 7) it is documented that the recirculation length of the wake, a quantity which is shown to be sensitive to buoyancy effects, does not differ significantly from passive scalar transport when $Ri_T = 1 \cdot 10^{-3}$, which is a value significantly higher than what is investigated in the present manuscript.

The Richardson number due to variations in water vapor content is defined as

$$Ri_{n_v} = -\frac{1}{\left(\frac{\rho_p}{\rho_\infty} - 1\right)} \frac{M_w - M_d}{N_a \rho_\infty} (n_{v,p} - n_{v,\infty}), \tag{12}$$

where $N_A$ is the Avogadro constant, $M_w$ the molar mass of water and $M_d$ the mixture molar mass of dry air (Chouippe et al., 2019, eq. (45)). Using the temperatures stated in the previous paragraph and the vapor boundary conditions stated in the manuscript, we obtain

$$Ri_{n_v} \approx 5 \cdot 10^{-6}, \tag{13}$$

which is orders of magnitudes smaller than $Ri_T$. Please note that eq. (4) in the manuscript is incorrect and should read

$$n_v = e/k_b T. \tag{14}$$

This will be corrected in the revised version of the manuscript.

Fluctuations in $\mathbf{v}_p$ are only important in the context of this manuscript if they lead to modifications in the structure of the wake. The equations of motion for the spherical particle suggest that the time scale of particle acceleration is proportional to $\rho_p/\rho_\infty$ (Chouippe et al., 2019, eq. (8)). If this time scale is much larger than the observation time of interest, which in our case is $L_x/v_p$ with $L_x$ being the length of the simulation domain, the wake will have a structure similar to that of particle falling through a fluid at rest with constant velocity. In other words the structure of the wake is only altered if the particle changes its falling direction significantly during the observation time. In (Chouippe et al., 2019, § 3.3) it is reported that a freely falling particle with $\rho_p/\rho_\infty = 10$ already behaves very similar to a fixed particle ($\rho_p/\rho_\infty \to \infty$) e.g. in terms of centerline temperature evolution and half-width of the thermal wake. Since we assume a much larger density ratio of 600, it seems unlikely that fluctuations in particle velocity have an impact on the shape of the wake.

*Adjustments to the manuscript:* Equation (4) of the discussion paper will be corrected. A sentence will be added to the manuscript stating that the wake of a freely falling hydrometeor behaves similar to that of a fixed particle due to the high value of the density ratio.

**R2-Co2. In a deep convective cloud, the hydrometeors are falling through a turbulent environment. Can the authors comment about the role of turbulent fluctuations in the ambient? How would the volume of the supersaturated region change with turbulence intensity in the ambient? There are some heat transfer studies from a heated sphere in a turbulent environment (Bagchi and Kottam 2008, Phys of Fluids). Can this be extended to the current study? It might be worthwhile to briefly discuss this as a part of future**

**work.**

The study of Bagchi and Kottam (2008) is very helpful when the effect of ambient turbulence on the supersaturated volume and ice enhancement is discussed. The parameter range investigated in their work corresponds reasonably well to the scenario of a hydrometeor with a diameter of a few millimeter settling under atmospheric conditions, where the largest flow scales are expected to be $\mathcal{O}(100\text{m})$ and the smallest scales around $\mathcal{O}(1\text{mm})$ (Lehmann, Siebert, and Shaw, 2009).

We know from simple mixing parcel models (e.g. (Chouippe et al., 2019, fig. 17) or (Prabhakaran et al., 2020, fig. 4)) that the highest supersaturations occur in regions where the temperature of the mixture is roughly halfway between $T_\infty$ and $T_p$, i.e. $\tilde{T} \approx 0.5$ with $\tilde{T}$ being the non-dimensionalized temperature as introduced in the manuscript. In (Bagchi and Kottam, 2008, fig. 17) it is demonstrated that the centerline temperature in the wake decays significantly faster if the background flow is turbulent, especially when $\tilde{T} \lessapprox 0.4$. Therefore it is to be expected that supersaturation decays faster in the wake than it does for a uniform inflow, and thus, the supersaturated volume as well as the ice enhancement are most likely smaller if the ambient is turbulent.

However, Bagchi and Kottam (2008) also investigated the effect of turbulence on the heat and mass transfer coefficient. While the mean value of the Nusselt number remains mostly unaffected, strong fluctuations in its value can be observed. This presumably leads to a more intermittent behavior of the temperature and vapor fields. The role of intermittency on ice nucleation activity still needs to be investigated more thoroughly, especially when the distribution of aerosol particles is explicitly considered (a point which was suggested to be investigated as part of future work). If regions of strong supersaturation conincide with regions where AP are preferentially located, intermittency might promote ice nucleation as supersaturation and nucleation rate are non-linearly linked to the temperature/vapor fields.

*Adjustments to the manuscript:* A paragraph discussing the influence of turbulence will be added to the manuscript.

*Note:* This response is the same as the response to remark R1-Co2 raised by Alexei Korolev, due to the strong similarity of the remarks.

**R2-Co3. Can the authors comment on how the supersaturated volume would be affected in the presence of cloud droplets and ice particles in the ambient?**

If a second riming ice particle or warm droplet approaches the settling ice particles, their thermal/vapor wakes will interact, which probably leads to reduced heat and mass transfer as temperature/vapor gradients are dampened in the boundary layer. Therefore the supersaturated volume induced by two nearby hydrometeors is likely to be smaller than their sum.

If a cloud droplet or ice particle which is colder than the riming meteor enters the wake, e.g. a hydrometeor at ambient temperature, water vapor may be removed from the gas phase by diffusional growth of the secondary hydrometeor. This presumably leads to a depletion of supersaturation, and hence, the supersaturated volume decreases.

**3  Minor remarks**

**R2-Mi1. Excess supersaturation - notation difference between Eq. 15 and Fig 4 caption. Fig 4 caption has a "*" on top of "s".**

We thank the reviewer for pointing out this mistake, which will be corrected in the revised version of the manuscript.

**4  Figures**

[Figure]

Figure 1: Isosurfaces of supersaturation in the wake at $T_\infty = -15°$C. The value of the isocontour is $\tilde{s}_i^* = 0.02$, i.e. two percentage points higher than the ambient supersaturation. Two different wake regimes are depicted, which correspond to two different hydrometeor sizes in our framework. (a) axisymmetric regime at $Re = 75$, (b) chaotic regime at $Re = 600$.
Changelog:  changed ambient temperature to $T_\infty = -15°$C; changed isocontour threshold to $\tilde{s}_i^* = 0.02$; adapted caption accordingly

[Figure]

a) $Re = 75$    b) $Re = 600$

Figure 2: Contours of excess supersaturation in the wake, averaged over time and azimuthal direction at $T_\infty = -15°C$. (a) axisymmetric regime at $Re = 75$, (b) chaotic regime at $Re = 600$.
Changelog:  changed ambient temperature to $T_\infty = -15°C$; adapted caption accordingly

[Figure]

Figure 3: Volume of air where supersaturation w.r.t. ice exceeds a given threshold as a function of the threshold. The volume is normalized by the volume of the ice particle and four different ambient temperatures are shown: $T_\infty = -6°C$ (——), $T_\infty = -9°C$ (——), $T_\infty = -12°C$ (——). $T_\infty = -15°C$ (——). Solid lines correspond to $Re = 600$ (chaotic regime), while dashed lines show the data obtained for $Re = 75$ (axisymmetric regime).
Changelog:  changed ambient temperature range; added $T_\infty$ indicator; adapted caption accordingly

[Figure]

Figure 4: Volume of air where supersaturation w.r.t. liquid exceeds a given threshold as a function of the threshold. The volume is normalized by the volume of the ice particle and four different ambient temperatures are shown: $T_\infty = -6°C$ (——), $T_\infty = -9°C$ (——), $T_\infty = -12°C$ (——). $T_\infty = -15°C$ (——). Solid lines correspond to $Re = 600$ (chaotic regime), while dashed lines show the data obtained for $Re = 75$ (axisymmetric regime).
Changelog:  new figure

[Figure]

Figure 5: Contours of local ice enhancement factor in the wake, averaged over time and azimuthal direction at $T_\infty = -15°C$. (a) axisymmetric regime at $Re = 75$, (b) chaotic regime at $Re = 600$.
Changelog:  changed ambient temperature to $T_\infty$  =  $-15°C$; contour lines are now linearly spaced; ice enhancement computed according to deposition nucleation law provided by Meyers et al. (1992); adapted caption accordingly

[Figure]

Figure 6: Volume of air with supersaturation above a given threshold as a function of the ice enhancement factor. The volume is normalized by the volume of the ice particle and four different temperatures are shown: $T_\infty = -6°C$ (——), $T_\infty = -9°C$ (——), $T_\infty = -12°C$ (——). $T_\infty = -15°C$ (——). Solid lines correspond to $Re = 600$ (chaotic regime), while dashed lines show the data obtained for $Re = 75$ (axisymmetric regime).
Changelog: changed ambient temperature range; ice enhancement computed according to deposition nucleation law provided by Meyers et al. (1992); x-axis now linearly spaced; added $T_\infty$ indicator; adapted caption accordingly

[Figure]

Figure 7: Global ice enhancement factor as a function of cloud temperature. The inset shows the same data, but in semi-logarithmic scale.
Changelog: changed ambient temperature range; ice enhancement computed according to deposition nucleation law provided by Meyers et al. (1992); adapted caption accordingly

[Figure]

Figure 8: Limiting cases for the nucleation rate $j_{met}$. The swept-volume limited estimation based on the considerations of Prabhakaran et al. (2020) is shown for $C_{expo} = 10$ (——) with the shaded area depicting the values obtained for $1 < C_{expo} < 100$. The exposure-time limited estimation, which is directly linked to the ice enhancement factor defined in the manuscript, is shown for $\tau_{nucl} = 10$s (——) and the range $1\text{s} < \tau_{nucl} < 100\text{s}$ (shaded area).
Changelog:  new figure

**References**

Al-Naimi, R. and Saunders, C. P. R.: Measurements of Natural Deposition and Condensation-Freezing Ice Nuclei with a Continuous Flow Chamber, Atmospheric Environment (1967), 19, 1871–1882, https://doi.org/10.1016/0004-6981(85)90012-5, 1985.

Bagchi, P. and Kottam, K.: Effect of Freestream Isotropic Turbulence on Heat Transfer from a Sphere, Physics of Fluids, 20, 073 305, https://doi.org/10.1063/1.2963138, 2008.

Chouippe, A., Krayer, M., Uhlmann, M., Dušek, J., Kiselev, A., and Leisner, T.: Heat and Water Vapor Transfer in the Wake of a Falling Ice Sphere and Its Implication for Secondary Ice Formation in Clouds, New Journal of Physics, 21, 043 043, https://doi.org/10.1088/1367-2630/ab0a94, 2019.

Fletcher, N. H.: Size Effect in Heterogeneous Nucleation, The Journal of Chemical Physics, 29, 572–576, https://doi.org/10.1063/1.1744540, 1958.

Greenan, B. J. W. and List, R.: Experimental Closure of the Heat and Mass Transfer Theory of Spheroidal Hailstones, Journal of the Atmospheric Sciences, 52, 3797–3815, https://doi.org/10.1175/1520-0469(1995)052¡3797:ECOTHA¿2.0.CO;2, 1995.

Hoose, C. and Möhler, O.: Heterogeneous Ice Nucleation on Atmospheric Aerosols: A Review of Results from Laboratory Experiments, Atmospheric Chemistry and Physics, 12, 9817–9854, https://doi.org/10.5194/acp-12-9817-2012, 2012.

Lehmann, K., Siebert, H., and Shaw, R. A.: Homogeneous and Inhomogeneous Mixing in Cumulus Clouds: Dependence on Local Turbulence Structure, Journal of the Atmospheric Sciences, 66, 3641–3659, https://doi.org/10.1175/2009JAS3012.1, 2009.

Meyers, M. P., DeMott, P. J., and Cotton, W. R.: New Primary Ice-Nucleation Parameterizations in an Explicit Cloud Model, Journal of Applied Meteorology, 31, 708–721, https://doi.org/10.1175/1520-0450(1992)031¡0708:NPINPI¿2.0.CO;2, 1992.

Prabhakaran, P., Kinney, G., Cantrell, W., Shaw, R. A., and Bodenschatz, E.: High Supersaturation in the Wake of Falling Hydrometeors: Implications for Cloud Invigoration and Ice Nucleation, Geophysical Research Letters, 47, e2020GL088 055, https://doi.org/10.1029/2020GL088055, 2020.

Schaller, R. C. and Fukuta, N.: Ice Nucleation by Aerosol Particles: Experimental Studies Using a Wedge-Shaped Ice Thermal Diffusion Chamber, Journal of the Atmospheric Sciences, 36, 1788–1802, https://doi.org/10.1175/1520-0469(1979)036¡1788:INBAPE¿2.0.CO;2, 1979.